# Boltzmann-Aligned Inverse Folding Model as a Predictor of Mutational Effects on Protein-Protein Interactions

**Xiaoran Jiao**[1][*]   **Weian Mao**[1,2][*]   **Wengong Jin**[3]   **Peiyuan Yang**[1],
**Hao Chen**[1][†]   **Chunhua Shen**[1,4][†]

[1] Zhejiang University, China   [2] MIT, USA   [3] Northeastern University, USA   [4] Ant Group

## Abstract

Predicting the change in binding free energy ($\Delta\Delta G$) is crucial for understanding and modulating protein-protein interactions, which are critical in drug design. Due to the scarcity of experimental $\Delta\Delta G$ data, existing methods focus on pre-training, while neglecting the importance of alignment. In this work, we propose Boltzmann Alignment technique to transfer knowledge from pre-trained inverse folding models to prediction of $\Delta\Delta G$. We begin by analyzing the thermodynamic definition of $\Delta\Delta G$ and introducing the Boltzmann distribution to connect energy to the protein conformational distribution. However, the protein conformational distribution is intractable. Therefore, we employ Bayes' theorem to circumvent direct estimation and instead utilize the log-likelihood provided by protein inverse folding models for the estimation of $\Delta\Delta G$.

Compared to previous methods based on inverse folding, our method explicitly accounts for the unbound state of the protein complex in the $\Delta\Delta G$ thermodynamic cycle, introducing a physical inductive bias and achieving supervised and unsupervised state-of-the-art (SoTA) performance. Experimental results on SKEMPI v2 indicate that our method achieves Spearman coefficients of 0.3201 (unsupervised) and 0.5134 (supervised) on SKEMPI v2, significantly surpassing the previously reported SoTA results of 0.2632 and 0.4324, respectively. Furthermore, we demonstrate the capability of our method in binding energy prediction, protein-protein docking, and antibody optimization tasks.

Code is available at https://github.com/aim-uofa/BA-DDG

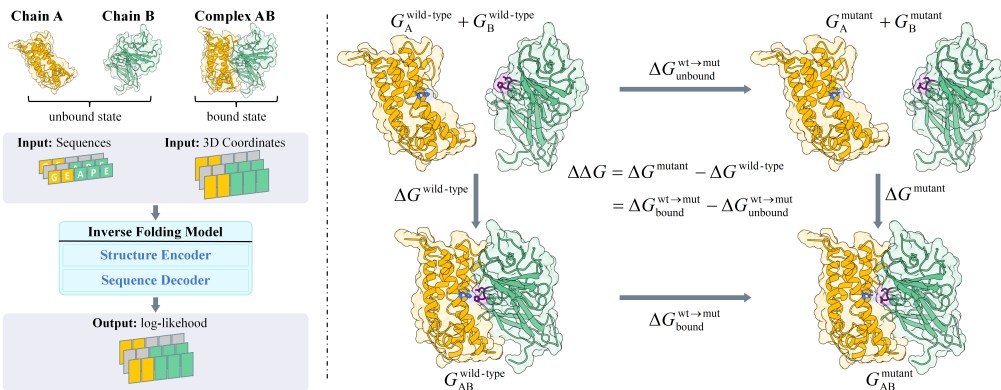

Figure 1: Overview of the Boltzmann Alignment technique. **Left**: inference with a protein inverse folding model. **Right**: illustration of thermodynamic cycle in the modulation of protein-protein interactions.

---

[*]XJ and WM contributed equally. Work was done when WM was visiting Zhejiang University. HC and CS are the corresponding authors.

# 1 INTRODUCTION

Protein–protein interactions (PPIs) are fundamental to the execution of diverse and essential biological functions across all organisms. High-fidelity computational modeling of protein-protein interactions is indispensable. The interaction properties of protein-protein binding can be quantitatively characterized by binding free energy ($\Delta G$), the difference in Gibbs free energy between the bound state and the unbound state. Predicting the change in binding free energy ($\Delta\Delta G$), also known as the mutational effect, is crucial to modulating protein-protein interactions. Accurate prediction of $\Delta\Delta G$ facilitates the identification of mutations that enhance or decrease the binding strength informs the efficient design of proteins, thus accelerating the development of therapeutic interventions and deepening our understanding of biological mechanisms.

Deep learning has demonstrated significant potential in protein modeling (Lin et al., 2022; Abramson et al., 2024), inspiring a paradigm shift in computational methods for predicting $\Delta\Delta G$. Traditional biophysics-based and statistics-based approaches (Park et al., 2016; Alford et al., 2017; Delgado et al., 2019) are increasingly being replaced by deep learning techniques (Shan et al., 2022a; Liu et al., 2023). Despite advancements, $\Delta\Delta G$ prediction is still hindered by *the scarcity of annotated experimental data*. Consequently, pre-training on extensive unlabeled data has emerged as a prevalent strategy for $\Delta\Delta G$ prediction (Luo et al., 2023; Wu et al., 2024). Recent studies observe that structure prediction models and inverse folding models implicitly capture the energy landscape (Bennett et al., 2023; Widatalla et al., 2024; Lu et al., 2024). Although effective, these pre-training-based approaches simply employ supervised fine-tuning (SFT) and neglect the importance of alignment. Since SFT may result in catastrophic forgetting of the general knowledge gained during unsupervised pre-training, there remains an opportunity for better knowledge transfer.

In other biological tasks, recent studies have adapted alignment techniques from large language models, such as direct preference optimization (DPO) (Rafailov et al., 2023), to integrate experimental fitness information into biological generative models (Zhou et al., 2024; Widatalla et al., 2024; Chennakesavalu et al., 2024; Liu et al., 2024). However, directly adopting these alignment techniques for $\Delta\Delta G$ prediction is inadequate, as they lack the physical inductive bias required for energy-related biological tasks.

In this work, we propose a technique named Boltzmann Alignment to transfer knowledge from pre-trained inverse folding models to $\Delta\Delta G$ prediction. We first analyze the thermodynamic definition of $\Delta\Delta G$ and introduce the Boltzmann distribution to connect energy with protein conformational distribution, thereby highlighting the potential of pre-trained probabilistic models. However, the protein conformational distribution is intractable. To address this, we employ Bayes' theorem to circumvent direct estimation and instead leverage the log-likelihood provided by protein inverse folding models for $\Delta\Delta G$ estimation. Our derivation provides a rational perspective on the previously observed high correlation between binding energy and log-likelihood from inverse folding models (Bennett et al., 2023; Widatalla et al., 2024; Shanker et al., 2024). Compared to previous inverse folding-based methods, our method explicitly considers the unbound state of the protein complex, enabling fine-tuning of inverse folding models in a manner consistent with statistical thermodynamics. Our contributions can be summarized as follows:

- We propose Boltzmann Alignment, which introduces physical inductive bias through Boltzmann distribution and thermodynamic cycle to transfer knowledge from pre-trained inverse folding models to $\Delta\Delta G$ prediction. We not only provide an alignment technique, but also offer insight into the high correlation previously observed between binding energy and log-likelihood in inverse folding models.

- Experimental results on SKEMPI v2 indicate that our method achieves Spearman coefficients of 0.3201 (unsupervised) and 0.5134 (supervised) on SKEMPI v2, significantly exceeding the previously reported SoTA values of 0.2632 and 0.4324, respectively. Further ablations demonstrate that our approach outperforms previous inverse folding-based methods and other alignment techniques. Additionally, we showcase the broader applicability of our method in binding energy prediction, protein-protein docking, and antibody optimization.

## 2 RELATED WORK

### 2.1 ALIGNMENT OF GENERATIVE MODELS

The prevailing pre-training approach for generative models, which focuses on maximizing the likelihood of training data, often falls short in aligning with specific user preferences. Consequently, there is growing interest in integrating task-specific information while maintaining the generative capabilities, especially in large language models (LLM). A commonly used alignment method is supervised fine-tuning (SFT), where models undergo additional training on curated datasets with specific annotations, utilizing a straightforward loss function. However, SFT poses the risk of overfitting to the fine-tuning dataset, which may result in catastrophic forgetting of the general knowledge gained during unsupervised pre-training. Much of the current alignment paradigm relies on reinforcement learning from human feedback (RLHF) (Ouyang et al., 2022). Within this framework, human preferences provided as pairwise rankings are used to train a reward model, which can then optimize a policy using proximal policy optimization (PPO) (Schulman et al., 2017). An alternative approach, known as direct preference optimization (DPO) (Rafailov et al., 2023), serves as a plug-and-play method that maximizes the likelihood of preferences. Numerous recent studies have expanded on the concepts of RLHF and DPO (An et al., 2023; Azar et al., 2023; Meng et al., 2024; Zhang et al., 2024).

Despite these advancements in LLM, developing effective methods to integrate experimental fitness information into biological generative models remains a crucial open question. Park et al. (2023) and Chennakesavalu et al. (2024) explore alignment in the context of inverse molecular design, enhancing the desired properties of generated compounds against a drug target. Liu et al. (2024) fine-tunes retrosynthetic planning models with a preference-based loss to improve the quality of generated synthetic routes. AbDPO (Zhou et al., 2024) introduces direct energy-based preference optimization to structure-sequence antibody co-design diffusion model, producing antibodies with low energy and high binding affinity. Mistani & Mysore (2024) incorporates DPO into protein language models to generate peptides with desirable physicochemical properties that bind to a given target protein. ProteinDPO (Widatalla et al., 2024) imbues structure-informed protein language model with biophysical information through DPO, enabling it to score stability and generate stable protein sequences. These efforts collectively underscore the potential of alignment techniques in advancing the performance of biological generative models.

### 2.2 PROTEIN-PROTEIN INTERACTION MODELING

Protein-protein interaction modeling has been extensively studied for decades, primarily concentrating on two key questions: where proteins interact, known as binding site prediction (Tubiana et al., 2022; Fang et al., 2023), and how they interact, addressed by protein-protein docking (Yan et al., 2020; Ketata et al., 2023). Recently, AlphaFold3 (Abramson et al., 2024) reaches almost 80% success rate of predicting general protein complex structures, effectively tackling both the "where" and "how" of protein-protein interactions. However, structure is not everything (Lowe, 2023); biomolecular interactions cannot be fully captured by static structures alone. A comprehensive understanding requires appreciating the dynamic processes of association and dissociation between protein partners (Lu et al., 2024), which are quantified by measures such as the binding affinity (Kd) or binding free energy ($\Delta G$). Predicting the change in binding free energy ($\Delta\Delta G$) is thus crucial for modulating protein-protein interactions; it enables the identification of mutations that enhance or diminish binding strength, which is essential for efficient protein design. For example, in antibody discovery, accurately predicting $\Delta\Delta G$ is fundamental to antibody maturation.

A variety of methods have been developed to predict $\Delta\Delta G$. Traditional approaches can be categorized into two main types: biophysical and statistical methods. Biophysical methods (Park et al., 2016; Alford et al., 2017; Delgado et al., 2019) use energy calculations to model how proteins interact at the atomic level, but they often struggle with balancing speed and accuracy. Statistical methods (Geng et al., 2019; Zhang et al., 2020) depend on feature engineering, utilizing descriptors that capture geometric, physical, and evolutionary characteristics of proteins. Traditional approaches depend heavily on human expertise and struggle to accurately capture complex interactions between proteins. Recently, deep learning-based approaches have emerged. Despite advancements in end-to-end learning approaches (Shan et al., 2022a), $\Delta\Delta G$ prediction is still limited by the lack of annotated experimental data. As a result, pre-training on extensive unlabeled data has become a common

strategy for improving predictions, involving various pre-training proxy tasks such as protein inverse folding (Yang et al., 2020), side-chain modeling (Luo et al., 2023; Liu et al., 2023; Mo et al., 2024), and masked modeling (Wu et al., 2024). These in-depth explorations of pre-training proxy tasks not only establish a solid foundation but also highlight the urgent need for transferring acquired knowledge to accurately predict $\Delta\Delta G$.

# 3 METHOD

Based on the Boltzmann distribution and thermodynamic cycles, as illustrated on the right side of Figure 1, we first establish the connection between $\Delta\Delta G$ and protein sequence likelihoods, which we call Boltzmann Alignment (Sec. 3.1). Next, we present a method (Sec. 3.2) that integrates the inverse folding model into Boltzmann alignment. This method is named BA-Cycle and uses the inverse folding model to evaluate $\Delta\Delta G$ by predicting the likelihoods of protein sequences, as shown on the left side of Figure 1. Upon BA-Cycle, we introduce a method named BA-DDG (Sec. 3.3) to fine-tune the inverse folding model using $\Delta\Delta G$-labeled data, introducing an inductive bias from statistical thermodynamics.

## 3.1 BOLTZMANN ALIGNMENT

To simplify the explanation, we introduce the case where the complex consists of two chains, chain A and chain B; however, the following content can be extended to cases with more chains. The binding free energy ($\Delta G$) of the complex is the difference between the Gibbs free energy of the bound state, denoted as $G_{\text{bnd}}$, and the Gibbs free energy of the unbound state, denoted as $G_{\text{unbnd}}$. Furthermore, using the Boltzmann distribution (Atkins et al., 2023), we can express the binding free energy ($\Delta G$) in terms of the probabilities of the two chains being in the bound state, $p_{\text{bnd}}$, and in the unbound state, $p_{\text{unbnd}}$, with the partition function eliminated by the two-state comparison:

$$\Delta G = G_{\text{bnd}} - G_{\text{unbnd}} = -k_{\text{B}}T \cdot (\log p_{\text{bnd}} - \log p_{\text{unbnd}}) \tag{1}$$

where $k_{\text{B}}$ is the Boltzmann constant and $T$ is the thermodynamic temperature. Specifically, the probabilities $p_{\text{bnd}}$ and $p_{\text{unbnd}}$ represent the probabilities of the protein complex structure being in the bound conformation $\mathcal{X}_{\text{bnd}}$ and the unbound conformation $\mathcal{X}_{\text{unbnd}}$, respectively, when the protein complex sequence $\mathcal{S}_{\text{AB}}$ is given. Therefore, $p_{\text{bnd}}$ and $p_{\text{unbnd}}$ can be expressed as the conditional probabilities $p(\mathcal{X}_{\text{bnd}} \mid \mathcal{S}_{\text{AB}})$ and $p(\mathcal{X}_{\text{unbnd}} \mid \mathcal{S}_{\text{AB}})$, where we simplify and approximate $\mathcal{X}_{\text{bnd}}$ and $\mathcal{X}_{\text{unbnd}}$ as the backbone structure of the protein complex. Substituting these conditional probabilities into Eq. 1 gives:

$$\Delta G = -k_{\text{B}}T \cdot (\log p(\mathcal{X}_{\text{bnd}} \mid \mathcal{S}_{\text{AB}}) - \log p(\mathcal{X}_{\text{unbnd}} \mid \mathcal{S}_{\text{AB}})) \tag{2}$$

Estimating $p(\mathcal{X} \mid \mathcal{S})$ poses several challenges. First, most current protein structure prediction models typically predict a single conformation and are not inherently probabilistic. Although these models provide uncertainty metrics such as pLDDT, pAE, and pTM (Abramson et al., 2024), and while these metrics have been observed to correlate with binding free energy (Lu et al., 2024), they do not offer a true probabilistic interpretation. Second, although recent efforts (Jing et al., 2023; 2024; Abramson et al., 2024) aim to introduce probabilistic models, specifically diffusion models, these neural networks learn to estimate the score $\nabla_{\mathcal{X}} \log p(\mathcal{X} \mid \mathcal{S})$ rather than $p(\mathcal{X} \mid \mathcal{S})$. Moreover, due to the relatively low sample efficiency of diffusion models, directly estimating $p(\mathcal{X} \mid \mathcal{S})$ remains intractable. Therefore, we use Bayes' theorem to circumvent direct estimation:

$$p(\mathcal{X} \mid \mathcal{S}) = \frac{p(\mathcal{S} \mid \mathcal{X}) \cdot p(\mathcal{X})}{p(\mathcal{S})} \tag{3}$$

Substituting this Bayes' theorem equation into Eq. 2 and eliminating the protein complex sequence probability $p(\mathcal{S}_{\text{AB}})$ gives:

$$\Delta G = -k_{\text{B}}T \cdot \left( \log \frac{p(\mathcal{S}_{\text{AB}} \mid \mathcal{X}_{\text{bnd}}) \cdot p(\mathcal{X}_{\text{bnd}})}{p(\mathcal{S}_{\text{AB}})} - \log \frac{p(\mathcal{S}_{\text{AB}} \mid \mathcal{X}_{\text{unbnd}}) \cdot p(\mathcal{X}_{\text{unbnd}})}{p(\mathcal{S}_{\text{AB}})} \right) \tag{4}$$

$$= -k_{\text{B}}T \cdot \log \frac{p(\mathcal{S}_{\text{AB}} \mid \mathcal{X}_{\text{bnd}}) \cdot p(\mathcal{X}_{\text{bnd}})}{p(\mathcal{S}_{\text{AB}} \mid \mathcal{X}_{\text{unbnd}}) \cdot p(\mathcal{X}_{\text{unbnd}})} \tag{5}$$

At this point, we have established the connection between the conditional probability of the protein sequence $p(\mathcal{S} \mid \mathcal{X})$ and $\Delta G$. Next, based on this, we further analyze the connection between the conditional probability of the protein sequence $p(\mathcal{S} \mid \mathcal{X})$ and $\Delta\Delta G$. Specifically, $\Delta\Delta G$ represents the difference between the $\Delta G$ of the wild type and the $\Delta G$ of the mutant type. The wild type and mutant type refer to the unmutated and mutated protein complexes, respectively. To distinguish between these two types, we denote them by adding "mut" and "wt" in the upper right corner of the notation; for example, $\Delta G^{\text{mut}}$ represents the $\Delta G$ of the mutant type. With this notation, we can express $\Delta\Delta G$ based on Eq. 5 as follows:

$$\Delta\Delta G = \Delta G^{\text{mut}} - \Delta G^{\text{wt}} \tag{6}$$

$$= -k_{\text{B}}T \cdot \left( \log \frac{p(\mathcal{S}_{\text{AB}}^{\text{mut}} \mid \mathcal{X}_{\text{bnd}}^{\text{mut}}) \cdot p(\mathcal{X}_{\text{bnd}}^{\text{mut}})}{p(\mathcal{S}_{\text{AB}}^{\text{mut}} \mid \mathcal{X}_{\text{unbnd}}^{\text{mut}}) \cdot p(\mathcal{X}_{\text{unbnd}}^{\text{mut}})} - \log \frac{p(\mathcal{S}_{\text{AB}}^{\text{wt}} \mid \mathcal{X}_{\text{bnd}}^{\text{wt}}) \cdot p(\mathcal{X}_{\text{bnd}}^{\text{wt}})}{p(\mathcal{S}_{\text{AB}}^{\text{wt}} \mid \mathcal{X}_{\text{unbnd}}^{\text{wt}}) \cdot p(\mathcal{X}_{\text{unbnd}}^{\text{wt}})} \right) \tag{7}$$

The backbone structure of the mutant type, $\mathcal{X}^{\text{mut}}$, is actually unknown. To address this issue, we introduce the assumption used in previous methods that the protein backbone structure remains unchanged before and after mutation. Therefore, we can approximate $\mathcal{X}_{\text{unbnd}}^{\text{wt}}$ to be equal to $\mathcal{X}_{\text{unbnd}}^{\text{mut}}$ and $\mathcal{X}_{\text{bnd}}^{\text{wt}}$ to be equal to $\mathcal{X}_{\text{bnd}}^{\text{mut}}$. Based on this assumption, we can eliminate the probability terms in Eq. 7:

$$\Delta\Delta G = -k_{\text{B}}T \cdot \left( \log \frac{p(\mathcal{S}_{\text{AB}}^{\text{mut}} \mid \mathcal{X}_{\text{bnd}}^{\text{mut}})}{p(\mathcal{S}_{\text{AB}}^{\text{mut}} \mid \mathcal{X}_{\text{unbnd}}^{\text{mut}})} - \log \frac{p(\mathcal{S}_{\text{AB}}^{\text{wt}} \mid \mathcal{X}_{\text{bnd}}^{\text{wt}})}{p(\mathcal{S}_{\text{AB}}^{\text{wt}} \mid \mathcal{X}_{\text{unbnd}}^{\text{wt}})} \right) \tag{8}$$

## 3.2 Probability Estimation

Eq. 8 transforms $\Delta\Delta G$ into the likelihoods of protein sequences $p(\mathcal{S}_{\text{AB}} \mid \mathcal{X})$. Here, we demonstrate the method of estimating the sequence probabilities of the bound state, $p(\mathcal{S}_{\text{AB}} \mid \mathcal{X}_{\text{bnd}})$, and the unbound state, $p(\mathcal{S}_{\text{AB}} \mid \mathcal{X}_{\text{unbnd}})$, using the inverse folding model. Since we assume that the backbone structure remains unchanged before and after mutation, i.e., $\mathcal{X}^{\text{wt}} = \mathcal{X}^{\text{mut}}$, the evaluation method for both the wild type and mutant type is identical, and no additional discussion is required.

**The bound state.** In our task, the backbone structure $\mathcal{X}_{\text{bnd}}$ of the bound state is usually known. Therefore, the inverse folding model can directly evaluate the probability $p(\mathcal{S}_{\text{AB}} \mid \mathcal{X}_{\text{bnd}})$. We denote the probability predicted by the inverse folding model as $p_\theta(\mathcal{S}_{\text{AB}} \mid \mathcal{X}_{\text{bnd}})$, where $\theta$ is the model parameter.

**The unbound state.** The unbound state is typically not explicitly provided in known backbone structure; therefore, we propose an alternative probability estimation method. The unbound state approximately represents the scenario where chains A and B of the complex are relatively far apart, with minimal interactions between them. Thus, a reasonable estimation approach is to independently evaluate these two chains using the inverse folding model. We denote the structure and sequence of chain A as $\mathcal{X}_{\text{A}}$ and $\mathcal{S}_{\text{A}}$, and those of chain B as $\mathcal{X}_{\text{B}}$ and $\mathcal{S}_{\text{B}}$. The estimation method for $p(\mathcal{S}_{\text{AB}} \mid \mathcal{X}_{\text{unbnd}})$ can be expressed as:

$$p(\mathcal{S}_{\text{AB}} \mid \mathcal{X}_{\text{unbnd}}) \approx p_\theta(\mathcal{S}_{\text{A}} \mid \mathcal{X}_{\text{A}}) \cdot p_\theta(\mathcal{S}_{\text{B}} \mid \mathcal{X}_{\text{B}}) \tag{9}$$

where $p_\theta(\mathcal{S}_{\text{A}} \mid \mathcal{X}_{\text{A}})$ and $p_\theta(\mathcal{S}_{\text{B}} \mid \mathcal{X}_{\text{B}})$ are the probabilities predicted by the inverse folding model.

**Unsupervised estimation of $\Delta\Delta G$.** Based on the estimation method described above, we can utilize a pretrained inverse folding model, ProteinMPNN (Dauparas et al., 2022), to achieve unsupervised evaluation of $\Delta\Delta G$. We name this approach BA-Cycle, which can be expressed as:

$$\widehat{\Delta\Delta G} = -k_{\text{B}}T \cdot \left( \log \frac{p_\theta(\mathcal{S}_{\text{AB}}^{\text{mut}} \mid \mathcal{X}_{\text{bnd}}^{\text{mut}})}{p_\theta(\mathcal{S}_{\text{A}}^{\text{mut}} \mid \mathcal{X}_{\text{A}}^{\text{mut}}) \cdot p_\theta(\mathcal{S}_{\text{B}}^{\text{mut}} \mid \mathcal{X}_{\text{B}}^{\text{mut}})} - \log \frac{p_\theta(\mathcal{S}_{\text{AB}}^{\text{wt}} \mid \mathcal{X}_{\text{bnd}}^{\text{wt}})}{p_\theta(\mathcal{S}_{\text{A}}^{\text{wt}} \mid \mathcal{X}_{\text{A}}^{\text{wt}}) \cdot p_\theta(\mathcal{S}_{\text{B}}^{\text{wt}} \mid \mathcal{X}_{\text{B}}^{\text{wt}})} \right) \tag{10}$$

**Comparison with previous work.** Previous work (Bennett et al., 2023; Cagiada et al., 2024; Widatalla et al., 2024; Shanker et al., 2024) attempts to use protein inverse folding models to predict $\Delta\Delta G$. However, these studies do not explicitly account for the unbound state $p_{\text{unbnd}}$ in the thermodynamic cycle. We denote these methods as $\widehat{\Delta\Delta G}_{\text{Prev}}$, summarized as:

$$\widehat{\Delta\Delta G}_{\text{Prev}} = -k_{\text{B}}T \cdot \left( \log p_\theta(\mathcal{S}_{\text{AB}}^{\text{mut}} \mid \mathcal{X}_{\text{bnd}}^{\text{mut}}) - \log p_\theta(\mathcal{S}_{\text{AB}}^{\text{wt}} \mid \mathcal{X}_{\text{bnd}}^{\text{wt}}) \right) \tag{11}$$

## 3.3 Supervision

We propose BA-DDG, a method that leverages $\Delta\Delta G$-labeled data to fine-tune BA-Cycle through Boltzmann Alignment. BA-DDG employs a forward process identical to that of BA-Cycle. During training, the parameters $\theta$ of the inverse folding model and $k_{\mathrm{B}}T$ in Eq. 10 are treated as learnable parameters that undergo optimization. The objective of BA-DDG is to minimize the discrepancy between the ground truth $\Delta\Delta G$ and the predicted $\widehat{\Delta\Delta G}$, while maintaining the distribution of the original pre-trained model, denoted as the reference model distribution $p_{\mathrm{ref}}(\mathcal{S} \mid \mathcal{X})$. The corresponding loss $\mathcal{L}_{\mathrm{Boltzmann}}$ can be expressed as:

$$\mathcal{L}_{\mathrm{Boltzmann}} = \underbrace{\|\Delta\Delta G - \widehat{\Delta\Delta G}\|}_{\text{minimize predicted error}} - \underbrace{\beta D_{\mathrm{KL}}(p_\theta(\mathcal{S} \mid \mathcal{X}) \| p_{\mathrm{ref}}(\mathcal{S} \mid \mathcal{X}))}_{\text{distributional penalty}} \tag{12}$$

where $\beta$ is a hyperparameter that controls the strength of the distributional penalty.

## 4 Experiments

In this section, we investigate three key questions through a series of experiments: (1) whether the physics-informed alignment technique we propose can achieve state-of-the-art (SoTA) accuracy on the SKEMPI v2 dataset under both unsupervised and supervised settings (Sec. 4.2); (2) whether the introduced thermodynamic cycle is indeed effective and has advantages over traditional SFT and DPO methods; (3) whether predicted structures can effectively replace crystal structures as inputs in our method (Sec. 4.3). Before addressing these questions, we first introduce the benchmarks and baselines (Sec. 4.1). Additionally, we explore potential applications of our method, taking binding energy prediction, protein-protein docking and antibody optimization as examples (Sec. 4.4).

## 4.1 Benchmark

**Dataset.** The SKEMPI v2 dataset (Jankauskaitė et al., 2019), the extensive annotated mutation dataset for 348 protein complexes, includes 7,085 amino acid mutations along with changes in thermodynamic parameters and kinetic rate constants. Despite lacking structures of the mutated complexes, it serves as the most well-established benchmark for $\Delta\Delta G$ prediction. To prevent data leakage, we follow Luo et al. (2023) and Wu et al. (2024), first dividing the dataset into 3 folds by structure, ensuring each fold contains unique protein complexes. This division forms the basis of our 3-fold cross-validation process, where two folds are used for training and validation, and the third fold is reserved for testing. This approach yields 3 different sets of parameters and ensures that every data point in SKEMPI v2 is tested once.

**Evaluation metrics.** To thoroughly assess the performance of $\Delta\Delta G$ prediction, we utilize a total of 7 metrics. This includes 5 overall metrics: (1) Pearson correlation coefficient, (2) Spearman's rank correlation coefficient, (3) minimized RMSE (root mean squared error), (4) minimized MAE (mean absolute error), and (5) AUROC (area under the receiver operating characteristic). To calculate AUROC, binary classification labels are assigned to mutations based on the sign of the $\Delta\Delta G$ predictions. In practical settings, the correlation for individual protein complexes tends to be of greater importance. Consequently, we group mutations by their structural characteristics, calculate the Pearson and Spearman correlation coefficients for each group, and then report the average of these per-structure correlations as 2 additional metrics.

**Baselines.** We compare our $\Delta\Delta G$ predictors, BA-Cycle and BA-DDG, with state-of-the-art unsupervised and supervised methods, respectively. Unsupervised methods are categorized into three groups: (1) traditional empirical energy functions such as Rosetta Cartesian $\Delta\Delta G$ (Alford et al., 2017) and FoldX (Delgado et al., 2019); (2) sequence/evolution-based approaches, exemplified by ESM-1v (Meier et al., 2021), Position-Specific Scoring Matrix (PSSM), MSA Transformer (Rao et al., 2021), and Tranception (Notin et al., 2022); (3) structure-informed pre-training-based methods not trained on $\Delta\Delta G$ labels, such as ESM-IF (Hsu et al., 2022), MIF-$\Delta$logits (Yang et al., 2020), RDE-Linear (Luo et al., 2023), and a model pre-trained to predict residue B-factors that are then used to predict $\Delta\Delta G$. Supervised methods can be divided into 2 categories: (1) end-to-end learning models, including DDGPred (Shan et al., 2022a) and a model that utilizes a self-attention-based network (Jumper et al., 2021) as the encoder but employs an MLP to directly predict $\Delta\Delta G$ (End-to-End); (2)

Table 1: Mean results of 3-fold cross-validation on the SKEMPI v2 dataset. **Bold** and underline indicate the best metrics for methods trained and not trained on the $\Delta\Delta G$ labels, respectively.

| Supervision | Method | Per-Structure | | Overall | | | | |
|---|---|---|---|---|---|---|---|---|
| | | **Pearson** ↑ | **Spear.** ↑ | **Pearson** ↑ | **Spear.** ↑ | **RMSE** ↓ | **MAE** ↓ | **AUROC** ↑ |
| ✗ | Rosetta | 0.3284 | 0.2988 | 0.3113 | 0.3468 | 1.6173 | 1.1311 | 0.6562 |
| | FoldX | 0.3789 | 0.3693 | 0.3120 | 0.4071 | 1.9080 | 1.3089 | 0.6582 |
| | ESM-1v | 0.0073 | -0.0118 | 0.1921 | 0.1572 | 1.9609 | 1.3683 | 0.5414 |
| | PSSM | 0.0826 | 0.0822 | 0.0159 | 0.0666 | 1.9978 | 1.3895 | 0.5260 |
| | MSA Transformer | 0.1031 | 0.0868 | 0.1173 | 0.1313 | 1.9835 | 1.3816 | 0.5768 |
| | Tranception | 0.1348 | 0.1236 | 0.1141 | 0.1402 | 2.0382 | 1.3883 | 0.5885 |
| | B-factor | 0.2042 | 0.1686 | 0.2390 | 0.2625 | 2.0411 | 1.4402 | 0.6044 |
| | ESM-IF | 0.2241 | 0.2019 | 0.3194 | 0.2806 | 1.8860 | 1.2857 | 0.5899 |
| | MIF-$\Delta$logit | 0.1585 | 0.1166 | 0.2918 | 0.2192 | 1.9092 | 1.3301 | 0.5749 |
| | RDE-Linear | 0.2903 | 0.2632 | 0.4185 | 0.3514 | 1.7832 | 1.2159 | 0.6059 |
| | BA-Cycle | 0.3722 | 0.3201 | 0.4552 | 0.4097 | 1.8402 | 1.3026 | 0.6657 |
| ✓ | DDGPred | 0.3750 | 0.3407 | 0.6580 | 0.4687 | 1.4998 | 1.0821 | 0.6992 |
| | End-to-End | 0.3873 | 0.3587 | 0.6373 | 0.4882 | 1.6198 | 1.1761 | 0.7172 |
| | MIF-Network | 0.3965 | 0.3509 | 0.6523 | 0.5134 | 1.5932 | 1.1469 | 0.7329 |
| | RDE-Network | 0.4448 | 0.4010 | 0.6447 | 0.5584 | 1.5799 | 1.1123 | 0.7454 |
| | DiffAffinity | 0.4220 | 0.3970 | 0.6609 | 0.5560 | 1.5350 | 1.0930 | 0.7440 |
| | Prompt-DDG | 0.4712 | 0.4257 | 0.6772 | 0.5910 | 1.5207 | 1.0770 | 0.7568 |
| | ProMIM | 0.4640 | 0.4310 | 0.6720 | 0.5730 | 1.5160 | 1.0890 | 0.7600 |
| | Surface-VQMAE | 0.4694 | 0.4324 | 0.6482 | 0.5611 | 1.5876 | 1.1271 | 0.7469 |
| | BA-DDG | **0.5453** | **0.5134** | **0.7118** | **0.6346** | **1.4516** | **1.0151** | **0.7726** |

structure-informed pre-training-based methods fine-tuned on $\Delta\Delta G$ labels, including MIF-Network (Yang et al., 2020), RDE-Network (Luo et al., 2023), DiffAffinity (Liu et al., 2023), Prompt-DDG (Wu et al., 2024), ProMIM (Mo et al., 2024), and Surface-VQMAE (Wu & Li, 2024).

## 4.2 MAIN RESULTS

**Comparison with baselines.** According to Table 1, our BA-DDG outperforms all the baselines across all evaluation metrics. Notably, it demonstrates a significant improvement in per-structure correlations, highlighting its greater reliability for practical applications. BA-Cycle achieves comparable performance to empirical energy functions and surpasses all unsupervised learning baselines. Detailed results for single-point, multi-point, and all-point mutations are available in Table 6.

**Visualization for correlation analysis.** We present scatter plots of experimental and predicted $\Delta\Delta G$ for three representative methods in Fig. 2, along with their overall Pearson and Spearman correlation scores. Additionally, Fig. 3 shows the distribution of per-structure Pearson and Spearman correlation scores, as well as the average results across all structures. It is evident that BA-DDG outperforms other methods in both qualitative visualization and quantitative metrics.

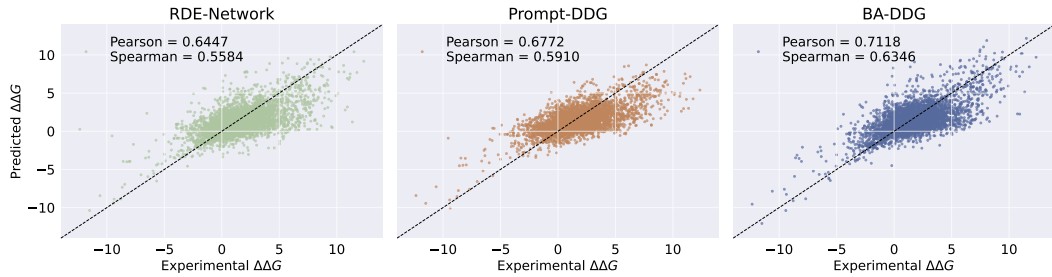

Figure 2: Comparison of correlations between experimental $\Delta\Delta G$ and $\Delta\Delta G$ predicted by three representative methods.

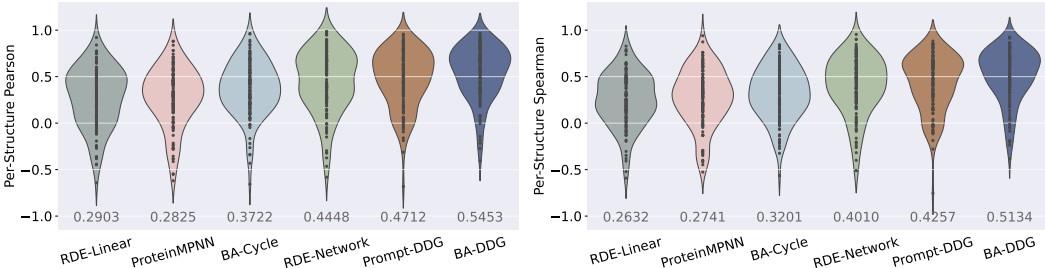

Figure 3: Distributions of per-structure Pearson correlation scores and Spearman correlation scores for six representative methods.

## 4.3 Ablation Study

**Ablation study on thermodynamic cycle.** As shown in Table 2, we evaluate the impact of thermodynamic cycle using 3-fold cross-validation on the SKEMPI v2 dataset. We compare our training-free version BA-Cycle to the basic usage of protein inverse folding models, as demonstrated in previous studies (Bennett et al., 2023; Cagiada et al., 2024; Widatalla et al., 2024). As described in Eq. 11, these studies do not account for thermodynamic cycle, as they consider only $p_{bnd}$ and neglect $p_{unbnd}$. **Disregarding thermodynamics significantly harms performance**, as BA-Cycle outperforms ESM-IF and the original ProteinMPNN.

Table 2: Ablation study on thermodynamic cycle. We report the mean results of 3-fold cross-validation on the SKEMPI v2 dataset. **Bold** indicate the best metrics.

| Method | Per-Structure | | Overall | | | | |
|---|---|---|---|---|---|---|---|
| | **Pearson ↑** | **Spear. ↑** | **Pearson ↑** | **Spear. ↑** | **RMSE ↓** | **MAE ↓** | **AUROC ↑** |
| ESM-IF | 0.2241 | 0.2019 | 0.3194 | 0.2806 | 1.8860 | **1.2857** | 0.5899 |
| ProteinMPNN | 0.2825 | 0.2741 | 0.2753 | 0.3112 | 1.9869 | 1.4048 | 0.6191 |
| BA-Cycle | **0.3722** | **0.3201** | **0.4552** | **0.4097** | **1.8402** | 1.3026 | **0.6657** |

**Ablation study on supervision methods.** As presented in Table 3, we evaluate the impact of various supervision methods. The training and inference settings are the same as in the main results. The implementation details of other alignment techniques can be found in Appendix A.1. **Boltzmann supervision yields the best performance** across all evaluation metrics, compared to other alignment techniques like SFT and DPO. These results clearly demonstrate the effectiveness of our proposed methods.

Table 3: Ablation study on supervision methods. We report the mean results of 3-fold cross-validation on the SKEMPI v2 dataset. **Bold** indicate the best metrics.

| Supervison | Per-Structure | | Overall | | | | |
|---|---|---|---|---|---|---|---|
| | **Pearson ↑** | **Spear. ↑** | **Pearson ↑** | **Spear. ↑** | **RMSE ↓** | **MAE ↓** | **AUROC ↑** |
| SFT | 0.5167 | 0.4769 | 0.7048 | 0.6240 | 1.4661 | 1.0346 | 0.7725 |
| DPO | 0.4212 | 0.3913 | 0.5397 | 0.4870 | 1.7399 | 1.2422 | 0.7036 |
| Boltzmann | **0.5453** | **0.5134** | **0.7118** | **0.6346** | **1.4516** | **1.0151** | **0.7726** |

**Inference with predicted structures.** The SKEMPI v2 benchmark includes wild-type crystal structures, which are also used by previous state-of-the-art methods such as Prompt-DDG (Wu et al., 2024) and RDE (Luo et al., 2023). However, crystal structures of mutants are often unknown, and using wild-type structures for mutants can introduce errors, especially with multiple mutation sites. In practical applications, such as antibody design, only sequence information may be available without a template crystal structure. To assess the usability of $\Delta\Delta G$ prediction models in various practical scenarios, we investigate whether our fine-tuned ProteinMPNN relies on crystal structures as inputs

to predict $\Delta\Delta G$, as presented in Table 4. **While the performance with AlphaFold3 predicted structures is slightly lower, the differences are minimal**, indicating that our method does not strictly require crystal structures when a reasonably accurate predicted structure is available.

Table 4: Comparison of inference with AlphaFold3 (Abramson et al., 2024) predicted structures and crystal structures. We report the mean results of 3-fold cross-validation on the common subset of Test Set 1 from SKEMPI as defined in SSIPe (Huang et al., 2019) and the SKEMPI dataset as employed in DSMBind (Jin et al., 2023). During the 3-fold cross-validation, models are trained on the SKEMPI v2 dataset using crystal structures as inputs and validated on the subset using AlphaFold3 predicted structures. Both the AlphaFold3 predicted structures and the dataset are provided by Lu et al. (2024).

| Structure | Per-Structure | | Overall | | | | |
|---|---|---|---|---|---|---|---|
| | Pearson ↑ | Spear. ↑ | Pearson ↑ | Spear. ↑ | RMSE ↓ | MAE ↓ | AUROC ↑ |
| Crystal | **0.8057** | 0.7322 | **0.8584** | **0.7946** | **1.3948** | **0.9713** | **0.8437** |
| AlphaFold3 | 0.8017 | **0.7459** | 0.8545 | 0.7821 | 1.4122 | 0.9817 | 0.8314 |

## 4.4 APPLICATION

### 4.4.1 BINDING ENERGY PREDICTION

Antibody design is vital for treating diseases such as cancer and infections. Although we lack a comprehensive explanation for canceling out $p(\mathcal{X}_{\text{bnd}})$ and $p(\mathcal{X}_{\text{unbnd}})$ in Eq. 5, BA-DDG can approximate $\Delta G$ using only inverse folding:

$$\widehat{\Delta G} \approx -k_{\text{B}}T \cdot (\log p_\theta(\mathcal{S}_{\text{AB}} \mid \mathcal{X}_{\text{AB}}) - (\log p_\theta(\mathcal{S}_{\text{A}} \mid \mathcal{X}_{\text{A}}) + \log p_\theta(\mathcal{S}_{\text{B}} \mid \mathcal{X}_{\text{B}}))) \quad (13)$$

We benchmark antibody-antigen binding energy predictors, with the experimental setup proposed in DSMBind (Jin et al., 2023). DSMBind is an unsupervised method trained on 3,416 antibody-antigen complexes from SAbDab (Schneider et al., 2021). SAbDab has over 3,500 non-redundant antibody-antigen complex structures, but only 566 of them have binding energy labels. Other methods involve pre-training on the entire binding energy dataset from SKEMPI, followed by fine-tuning on antibody-antigen data from SKEMPI. We evaluate these methods on a SAbDab subset of 566 complexes with binding energy labels. As depicted in Fig. 4 (left), BA-DDG achieves a Spearman correlation of 0.385, outperforming other methods.

### 4.4.2 RIGID PROTEIN-PROTEIN DOCKING

DiffDock-PP (Ketata et al., 2023) is a diffusion generative model specifically designed for rigid protein-protein docking. It learns to translate and rotate unbound protein structures into their bound

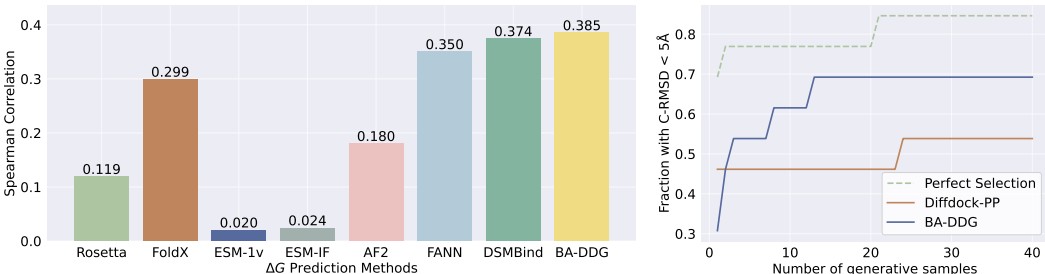

Figure 4: **Left**: The Spearman correlation between predicted and true binding free energy ($\Delta G$) on the SAbDab test set. **Right**: Performance of selection methods for rigid protein-protein docking as the number of generative samples increases. "Perfect Selection" shows the best possible performance with an ideal selection method. "Diffdock-PP" refers to the confidence model proposed by Diffdock-PP (Ketata et al., 2023). "BA-DDG" involves using our method to estimate $\Delta G$ for selection. Performance is assessed by calculating the fraction of 13 antibody complex generation tasks on the DIPS test set (Townshend et al., 2019) that achieve a C-RMSD of less than 5Å.

conformations. The docking process comprises two primary steps. First, the diffusion model generates samples. Then, a pre-trained confidence model evaluates these samples to identify the one with the highest confidence score, which is used as the final docking result. BA-DDG can serve as an approximate binding free energy ($\Delta G$) estimator and can be employed as a selection method in diffusion-based docking process. A lower $\Delta G$ indicates a more spontaneous and favorable transition from the unbound to the bound state, corresponding to a higher confidence score for the generative docking sample. By integrating BA-DDG into the diffusion-based docking process, we provide a robust method for selecting the most plausible docking conformation, as presented in Fig. 4 (right).

### 4.4.3 ANTIBODY OPTIMIZATION

**Antibody optimization against SARS-CoV-2.** Beyond serving as an energy scoring function, BA-DDG also acts as a structure-based protein designer, similar to ProteinMPNN. Consequently, we are curious **whether the ProteinMPNN fine-tuned for $\Delta\Delta G$ prediction tends to generate sequences with higher affinity**. Shan et al. (2022b) identify five single-point mutations in a human antibody targeting SARS-CoV-2 that significantly enhance its neutralization effectiveness. These mutations are selected from a pool of 494 possible single-point mutations within the heavy chain CDR region of the antibody. We calculate the perplexity differences and preference probabilities of these 494 possible single-point mutations relative to the wild-type sequence. Specifically, normalized per-amino acid perplexity is obtained by subtracting the average per-amino acid perplexity of all possible sequences from that of the mutant sequence. Preference probabilities are derived using the Bradley-Terry model (Bradley & Terry, 1952) applied to the predicted probabilities of wild-type and mutant sequences. The results, as presented in the Table 5, show that the fine-tuned BA-DDG generally achieves lower normalized perplexity and higher preference probabilities compared to the original ProteinMPNN, indicating a tendency to generate sequences with potentially higher binding affinity and improved neutralization effectiveness. Results on multi-point mutations, provided in the Table 7, further support this conclusion. This suggests that **fine-tuning for $\Delta\Delta G$ prediction can enhance the design of more effective antibody sequences**.

Table 5: Normalized per-amino acid perplexity and preference probabilities of the five favorable mutations sequence on the antibody against SARS-CoV-2 (Shan et al., 2022b) by original and fine-tuned ProteinMPNN.

| Metric | Method | TH31W | AH53F | NH57L | RH103M | LH104F | Average |
|---|---|---|---|---|---|---|---|
| Perplexity ↓ | ProteinMPNN | 6.018 | **0.179** | **0.401** | 2.895 | 2.521 | 2.403 |
| | BA-DDG | **1.979** | 0.429 | 0.809 | **2.416** | **0.992** | **1.325** |
| Preference ↑ | ProteinMPNN | 12.61% | 62.55% | **75.62%** | 13.93% | 36.20% | 42.47% |
| | BA-DDG | **37.24%** | **68.65%** | 56.51% | **25.10%** | **47.63%** | **47.03%** |

## 5 DISCUSSION AND CONCLUSION

We propose Boltzmann Alignment, a method that leverages physical inductive bias through the Boltzmann distribution and thermodynamic cycle to enhance $\Delta\Delta G$ prediction by transferring knowledge from pre-trained inverse folding models. Experiments on SKEMPI v2 yield Spearman coefficients of 0.3201 (unsupervised) and 0.5134 (supervised), significantly surpassing previous state-of-the-art values of 0.2632 and 0.4324. Additionally, we demonstrate the broader applicability of our approach in binding energy prediction, protein-protein docking, and antibody optimization.

Our method has certain limitations that we aim to address in future work. Firstly, our approach requires crystal structures or reliable predicted structures as input; in cases where mutant protein structures are unavailable, we assume that the mutant backbone structure is identical to that of the wild-type protein; we take parts of the complex backbone structure as the backbone structure of single proteins. This reliance on structural data may limit the applicability and effectiveness of our method. Moreover, our current model does not consider side-chain conformations, which may be essential for $\Delta\Delta G$ prediction. Incorporating side-chain flexibility could enhance predictive accuracy and provide deeper insights into protein-protein interactions.

The **positive societal impact** is that this work can assist in drug design and virtual screening potentially. No **negative societal impact** is perceived.

ACKNOWLEDGMENTS

This work was supported by the National Key R&D Program of China (NO.2022ZD0160101) and the National Natural Science Foundation of China (No. 62206244).

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

# A IMPLEMENTATION DETAILS

## A.1 OTHER ALIGNMENT TECHNIQUES

### A.1.1 SFT

The training objective of trivial SFT is as follows:

$$\mathcal{L}_{\text{SFT}} = \underbrace{\|\Delta\Delta G - \widehat{\Delta\Delta G}_{\text{Prev}}\|}_{\text{minimize predicted error}} - \underbrace{\beta D_{\text{KL}}(p_\theta(S \mid X)\|p_{\text{ref}}(S \mid X))}_{\text{distributional penalty}} \tag{14}$$

### A.1.2 DPO

We utilize weighted DPO algorithm proposed in Widatalla et al. (2024). The weighted DPO objective is simply minimization of the KL-divergence between the distribution parameterized by the generative model and numerical scores ($\Delta\Delta G$ labels).

## A.2 MODEL

In ProteinMPNN, part of the input protein sequence is to be designed, and the rest is fixed; decoding skips fixed regions but includes them in the context for designing the remaining positions during autoregressive decoding. When estimating $\Delta\Delta G$ with ProteinMPNN, we designate the mutation sites $S_{\text{D}}$ to be designed while fixing the rest of the sequence $S \setminus S_{\text{D}}$. Let $\pi = (s_1, s_2, \ldots, s_n)$ represent a randomly sampled decoding order, a permutation of the mutation sites $S_{\text{D}}$. The joint probability $p_\theta(S \mid X)$ can be expanded step by step into the product of model-predicted probabilities for each amino acid site using the conditional probability formula:

$$p_\theta(S \mid X) = p_\theta(S_{\text{D}} \mid X, S \setminus S_{\text{D}}) \tag{15}$$
$$= p_\theta(S_{\text{D}} \setminus \{s_1\} \mid X, S \setminus S_{\text{D}} \cup \{s_1\}) \cdot p_\theta(\{s_1\} \mid X, S \setminus S_{\text{D}}) \tag{16}$$
$$= \ldots$$

## A.3 TRAINING

### A.3.1 HYPER-PARAMETERS

We train the model over 20,000 iterations using the Adam optimizer, with a learning rate set at 0.0001 and a batch size of 2. The small batch size is because we need to include both the wild-type and mutant complexes along with their respective monomer structures in a single batch for each iteration. The model is insensitive to the loss weight parameter $\beta$, so it is typically set to a low value, such as 0.001, during training.

For the inverse folding model, we utilized the pre-trained ProteinMPNN, which includes 3 MPNN layers. It considers the 30 nearest neighbors to construct edges and has an embedding dimension of 256.

### A.3.2 HARDWARE

All our experiments are conducted on a computing cluster with CPUs of AMD EPYC 7763 64-Core of 3.52GHz and a single NVIDIA GeForce RTX 4090 24GB GPU.

## A.4 ASSUMPTION

**Independence approximation in the unbound state.**

The unbound state generally refers to a condition where no significant inter-chain interactions exist. While weak inter-chain interactions (e.g., transient or solvent-mediated interactions) may occasionally occur in this state, such interactions typically exert minimal influence on free energy calculations. This supports the common and effective assumption of treating the unbound state as independent probabilities for individual chains, a simplification that substantially reduces computational complexity while introducing negligible error.

Conversely, inter-chain interactions dominate in the bound state. To explicitly model these critical interactions, we calculate the bound state's probability by inputting the entire complex into Protein-MPNN. This approach retains all inter-chain interactions through their explicit representation as edges in the model's graph structure.

# B ADDITIONAL RESULTS

## B.1 BINDING ENERGY PREDICTION

We compare BA-DDG with several representative methods under single-point, multi-point, and all-point mutations on SKEMPI v2. The results reported in Table 6 demonstrate that BA-DDG achieves the best overall performance for single-point mutations and the best per-structure performance for multi-point mutations. Notably, BA-DDG performs better on single-point mutations compared to multi-point mutations, which may be related to the autoregressive decoding of ProteinMPNN. In our BA-DDG, when dealing with multi-point mutations, the probability estimation for a new site depends on the estimation of the previous site. This coupling might affect performance, presenting an opportunity for future improvements.

Table 6: Performance comparison under single-, multi- and all-point mutation on SKEMPI v2, where **bold** denotes the best metric under each setting.

| Method | Mutations | Per-Structure | | Overall | | | | |
|---|---|---|---|---|---|---|---|---|
| | | Pearson ↑ | Spear. ↑ | Pearson ↑ | Spear. ↑ | RMSE ↓ | MAE ↓ | AUROC ↑ |
| Rosetta | all | 0.3284 | 0.2988 | 0.3113 | 0.3468 | 1.6173 | 1.1311 | 0.6562 |
| | single | 0.3510 | 0.4180 | 0.3250 | 0.3670 | 1.1830 | 0.9870 | 0.6740 |
| | multiple | 0.1910 | 0.0830 | 0.1990 | 0.2300 | 2.6580 | 2.0240 | 0.6210 |
| FoldX | all | 0.3789 | 0.3693 | 0.3120 | 0.4071 | 1.9080 | 1.3089 | 0.6582 |
| | single | 0.3820 | 0.3600 | 0.3150 | 0.3610 | 1.6510 | 1.1460 | 0.6570 |
| | multiple | 0.3330 | 0.3400 | 0.2560 | 0.4180 | 2.6080 | 1.9260 | 0.7040 |
| DDGPred | all | 0.3750 | 0.3407 | 0.6580 | 0.4687 | 1.4998 | 1.0821 | 0.6992 |
| | single | 0.3711 | 0.3427 | 0.6515 | 0.4390 | 1.3285 | 0.9618 | 0.6858 |
| | multiple | 0.3912 | 0.3896 | 0.5938 | 0.5150 | 2.1813 | 1.6699 | 0.7590 |
| End-to-End | all | 0.3873 | 0.3587 | 0.6373 | 0.4882 | 1.6198 | 1.1761 | 0.7172 |
| | single | 0.3818 | 0.3426 | 0.6605 | 0.4594 | 1.3148 | 0.9569 | 0.7019 |
| | multiple | 0.4178 | 0.4034 | 0.5858 | 0.4942 | 2.1971 | 1.7087 | 0.7532 |
| RDE-Network | all | 0.4448 | 0.4010 | 0.6447 | 0.5584 | 1.5799 | 1.1123 | 0.7454 |
| | single | 0.4687 | 0.4333 | 0.6421 | 0.5271 | 1.3333 | 0.9392 | 0.7367 |
| | multiple | 0.4233 | 0.3926 | 0.6288 | 0.5900 | 2.0980 | 1.5747 | 0.7749 |
| DiffAffinity | all | 0.4220 | 0.3970 | 0.6609 | 0.5560 | 1.5350 | 1.0930 | 0.7440 |
| | single | 0.4290 | 0.4090 | 0.6720 | 0.5230 | 1.2880 | 0.9230 | 0.7330 |
| | multiple | 0.4140 | 0.3870 | 0.6500 | 0.6020 | 2.0510 | 1.5400 | 0.7840 |
| Prompt-DDG | all | 0.4712 | 0.4257 | 0.6772 | 0.5910 | 1.5207 | 1.0770 | 0.7568 |
| | single | 0.4736 | 0.4392 | 0.6596 | 0.5450 | 1.3072 | 0.9191 | 0.7355 |
| | multiple | 0.4448 | 0.3961 | **0.6780** | **0.6433** | 1.9831 | **1.4837** | 0.8187 |
| ProMIM | all | 0.4640 | 0.4310 | 0.6720 | 0.5730 | 1.5160 | 1.0890 | 0.7600 |
| | single | 0.4660 | 0.4390 | 0.6680 | 0.5340 | 1.2790 | 0.9240 | 0.7380 |
| | multiple | 0.4580 | 0.4250 | 0.6660 | 0.6140 | **1.9630** | 1.4910 | **0.8250** |
| BA-DDG | all | **0.5453** | **0.5134** | **0.7118** | **0.6346** | **1.4516** | **1.0151** | **0.7726** |
| | single | **0.5606** | **0.5192** | **0.7321** | **0.6157** | **1.1848** | **0.8409** | **0.7662** |
| | multiple | **0.4924** | **0.4959** | 0.6650 | 0.6293 | 2.0151 | 1.4944 | 0.7875 |

## B.2 ANTIBODY OPTIMIZATION

We provide results on multi-point mutations, which further support the conclusion that fine-tuning ProteinMPNN for $\Delta\Delta G$ prediction enhances the design of more effective antibody sequences. These findings confirm that the fine-tuned BA-DDG generally generates sequences with lower perplexity and higher preference probabilities, indicating higher binding affinity and improved neutralization effectiveness.

Table 7: Normalized per-amino acid perplexity and preference probabilities for the multi-point mutation sequences derived from multiple optimization rounds on the antibody against SARS-CoV-2 (Shan et al., 2022b), as predicted by the original and fine-tuned ProteinMPNN.

| Mutations | Perplexity ↓ | | Preference ↑ | |
|---|---|---|---|---|
| | BA-DDG | ProteinMPNN | BA-DDG | ProteinMPNN |
| RH103M,LH104F | 1.556 | 2.524 | 32.12% | 17.60% |
| TH30P,RH103M | 1.023 | 1.310 | 53.22% | 45.09% |
| TH31W,RH103M | 1.669 | 3.894 | 36.87% | 13.63% |
| AH39F,RH103M | 1.658 | 6.312 | 31.13% | 2.478% |
| NH55L,RH103M | 1.465 | 2.600 | 36.04% | 15.20% |
| NH57L,RH103M | 1.366 | 0.936 | 44.32% | 57.68% |
| TH30P,RH103M,LH104F | 1.151 | 2.181 | 45.21% | 22.29% |
| TH31W,RH103M,LH104F | 1.442 | 2.941 | 39.40% | 22.71% |
| TH30P,TH31W,RH103M | 1.203 | 2.606 | 45.74% | 22.44% |
| AH39F,RH103M,LH104F | 1.300 | 4.621 | 37.94% | 4.705% |
| AH39F,NH55L,RH103M | 1.500 | 4.929 | 34.63% | 3.416% |
| NH55L,NH57L,RH103M | 1.210 | 1.403 | 42.93% | 36.75% |
| AH39F,NH57L,RH103M | 1.409 | 2.329 | 38.18% | 15.82% |
| TH30P,AH39F,RH103M | 1.136 | 2.841 | 43.07% | 10.82% |
| TH31W,AH39F,RH103M | 1.606 | 6.742 | 34.08% | 2.438% |
| TH31W,NH55L,RH103M | 1.419 | 3.302 | 39.33% | 18.19% |
| TH30P,NH57L,RH103M | 1.012 | 0.948 | 52.97% | 51.40% |
| TH31W,NH57L,RH103M | 1.465 | 2.556 | 40.62% | 27.91% |
| AH39F,TH30P,RH103M,LH104F | 1.081 | 3.036 | 44.56% | 7.513% |
| **Average** | **1.351** | 3.053 | **40.65%** | 20.95% |

