# OpenReview forum: "Boltzmann-Aligned Inverse Folding Model as a Predictor of Mutational Effects on Protein-Protein Interactions"
_ICLR.cc/2025/Conference — ICLR 2025 Spotlight_

### Official Review · Reviewer_5mfi · 2024-10-29

**Soundness:** 3
**Presentation:** 3
**Contribution:** 4
**Rating:** 8
**Confidence:** 4

**Summary:**

The paper proposes a novel and interesting method that leverages Boltzmann Alignment to establish a more sophisticated connection between inverse folding tasks and binding free energy (ΔΔG) prediction. Despite relying on somewhat idealized approximations, the method leverages a thermodynamic cycle and the principles of the Boltzmann distribution to incorporate both bound and unbound states of the protein complex, offering a physically motivated alignment technique that addresses limitations of traditional pre-trained inverse folding models.  Through evaluations on the selected SKEMPI v2 dataset, the paper demonstrates that the method achieves significant improvements over existing approaches in both supervised and unsupervised settings.  Overall, the paper provides a new perspective for integrating domain knowledge into alignment techniques.

**Strengths:**

1) The paper is well-written.

2) The approach achieves state-of-the-art results compared to baselines.

3) The paper effectively incorporates domain-specific priors, specifically leverages Boltzmann principles to connect energy with probability, along with the assumption that unbound states can be approximated as independent probabilities for each chain. Though this assumption is idealized, it still is an interesting approach that introduces a motivated alignment.

4) The paper demonstrates practical applicability on binding energy prediction, protein-protein docking and antibody optimization tasks.

**Weaknesses:**

1) The method makes strong assumptions, particularly in treating the unbound state as independent probabilities for each chain, which oversimplifies the inter-chain interactions. Additionally, the predictions of inverse folding model’s do not fully align with the predicted probabilities (not the ground truth), introducing biases when single-chain probabilities are combined to approximate complex states. This approximation may result in significant deviations in some situations.

2) The method has only been validated on a limited dataset, which lacks sufficient evidence to demonstrate the generalizability of the approach and its underlying assumptions.

3) It seems that the paper lacks sufficient discussion on how improvements in the inverse folding model would translate to gains in this approach.  It’s also unclear which specific module in this sophisticated approach contributes most significantly to the information gain.

**Questions:**

1) Could the authors explain the reasons for the relatively low MAE reported in Table 2?

2) Could the authors clarify how they determined the unbound state approximation for independent chain probabilities (Equation 9)?  The paper provides some explanation on this point, but it lacks sufficient detail.

3) The transformation between energy and probability, along with the approximation of unbound state probabilities, partially explains the high correlation between the inverse folding task and ΔΔG prediction. However, within the neural network, the sophisticated design of model doesn’t appear to provide additional information gain.

Given the strengths of this paper, I would consider raising my score if the authors can address my concerns.

---

> ### Author Response · Authors · 2024-11-22
> **Response to Reviewer 5mfi- Part (1/2)**
>
> Dear Reviewer:
>
> Thank you for your thorough review and constructive comments! We sincerely hope our responses fully address your questions.
>
> > **W1.1**: The method makes strong assumptions, particularly in **treating the unbound state as independent probabilities for each chain**, which oversimplifies the inter-chain interactions.
>
> **A**: Thanks for your constructive comment. The unbound state typically refers to a condition with no significant inter-chain interactions. We acknowledge that weak inter-chain interactions, such as transient and solvent-mediated interactions, may exist in the unbound state. However, these interactions generally have minimal impact on free energy, making it a common and effective assumption to treat the unbound state as independent probabilities for each chain, thereby simplifying the problem with negligible cost.
> In contrast, inter-chain interactions are much stronger in the bound state. To account for this, we compute its probability by inputting the entire complex into ProteinMPNN, preserving all inter-chain interactions represented as edges in the model’s graph structure.
>
> > **W1.2**: Additionally, **the predictions of inverse folding model’s do not fully align with the predicted probabilities (not the ground truth)**, introducing biases when single-chain probabilities are combined to approximate complex states. This approximation may result in significant deviations in some situations.
>
> **A**: Thanks for your insightful comment. We agree that, although the pretrained inverse folding model we use is probabilistic, its predicted probabilities do not reflect the ground truth because it is simply trained using a cross-entropy loss on biologist-resolved structure-sequence pairs. However, our proposed Boltzmann Alignment establishes a bidirectional connection between the inverse folding model's predicted probabilities and the $\Delta\Delta G$ values, enabling mutual enhancement. Experimental results demonstrate that this approach not only enables the use of the inverse folding pretrained model to predict $\Delta\Delta G$ but also allows the fine-tuned inverse folding model to generate sequences with higher affinity (Section 4.4.3).
>
> We are not sure if we have correctly understood and addressed your question here. If you have any further questions, please feel free to let us know.
>
> >**W2**: The method has only been validated on a **limited dataset**, which lacks sufficient evidence to demonstrate the generalizability of the approach and its underlying assumptions.
>
> **A**: Thanks for pointing this out. (1) SKEMPIv2 remains the **largest and most well-established** benchmark available for $\Delta\Delta G$  prediction. (2) Due to the high cost of measurements and the lack of high-throughput techniques, experimentally determined $\Delta\Delta G$ data is limited in availability. (3) We also hope to see more public datasets in this field. Recent efforts, such as AlphaSeq, are promising, though their whole dataset is **not yet publicly available**.
>
> >**W3**: It seems that the paper lacks sufficient discussion on how improvements in the inverse folding model would translate to gains in this approach. It’s also unclear which specific module in this sophisticated approach contributes most significantly to the information gain.
>
> **A**: Thanks. Our main contribution lies in incorporating physical inductive bias to more effectively leverage the inverse folding model for $\Delta\Delta G$ prediction.  We do not introduce any sophisticated model design, except for a single learnable parameter, $k_B T$. Therefore, in our opinion, we consider the ablation study (on thermodynamic cycle and supervision method) in Section 4.3 to be clear and sufficient. However, **we would appreciate any further suggestions or specific ablation experiments you think could enhance the clarity of our results**.

---

> ### Author Response · Authors · 2024-11-22
> **Response to Reviewer 5mfi- Part (2/2)**
>
> >**Q1**: Could the authors explain the reasons for the relatively low MAE reported in Table 2?
>
> **A**: As shown in Equation 10, the unsupervised estimation of $\Delta\Delta G$ relies not only on the conditional probabilities modeled by the inverse folding model but also on $k_B T$ (the product of Boltzmann constant and thermodynamic temperature). In the supervised version, we treat $k_B T$ as a learnable parameter, whereas in the unsupervised version, we simply set $k_B T = 1$. While this unknown scaling factor does not impact correlation-based metrics, it does affect absolute metrics like MAE, which may account for the relatively low MAE reported in Table 2.
>
>
> >**Q2**: Could the authors clarify how they determined the unbound state approximation for independent chain probabilities (Equation 9)? The paper provides some explanation on this point, but it lacks sufficient detail.
>
> **A**: Thanks and addressed. As illustrated in Fig.1-Left, for a single chain (A) extracted from a complex (AB), we mask both the sequence and structure of the complex AB according to chain A's mask. The resulting masked information is then input into the protein inverse folding model, ProteinMPNN.
> For the unbound state, each chain is evaluated independently—chain A is processed in isolation, disregarding chain B, and vice versa. Conversely, for the bound state, chains A and B are processed together in ProteinMPNN, preserving all inter-chain interactions (represented as inter-chain edges in ProteinMPNN's graph representation).
>
> >**Q3**: The transformation between energy and probability, along with the approximation of unbound state probabilities, partially explains the high correlation between the inverse folding task and ΔΔG prediction. However, within the neural network, the sophisticated design of model doesn’t appear to provide additional information gain.
>
> **A**: Thanks for your comment. In our opinion, we do not introduce any sophisticated model design; the only additional learnable parameter compared to the inverse folding model is a scalar $k_B T$ (the product of the Boltzmann constant and thermodynamic temperature). Our main contribution lies in incorporating physical inductive bias to more effectively leverage the inverse folding model for $\Delta\Delta G$ prediction. By incorporating the thermodynamic cycle and Boltzmann distribution, we can achieve better performance than previous $\Delta\Delta G$ predictors (including previous inverse-folding-based methods).

---

> > ### Comment · Reviewer_5mfi · 2024-11-27
> >
> > Considering your response, as well as the replies provided to other reviewers, my concerns have been addressed. Although I believe the assumptions made in this work are too strong, I also acknowledge the novelty of this work. I will raise my score for this paper.

---

> > > ### Author Response · Authors · 2024-11-30
> > >
> > > Your insightful comments and kind support are greatly appreciated! As the discussion period ends on Dec. 2nd, we welcome any further questions you may have.

---

### Official Review · Reviewer_Mp9v · 2024-11-02

**Soundness:** 3
**Presentation:** 3
**Contribution:** 3
**Rating:** 6
**Confidence:** 3

**Summary:**

The paper presents Boltzmann Alignment, a novel approach for predicting changes in protein-protein binding free energy due to mutations. It utilizes pre-trained inverse folding models and introduces physical inductive bias through the Boltzmann distribution.

**Strengths:**

- The integration of Boltzmann distribution within the framework to enhance ∆∆G prediction is innovative and addresses a significant challenge in computational biology.
- The authors effectively showcase the versatility of their method in binding energy prediction, protein-protein docking, and antibody optimization, indicating its broad applicability.
- The paper is well-organized, and the results are presented clearly, making it easy for readers to follow the methodology and findings.

**Weaknesses:**

I haven't identified any significant weaknesses.

**Questions:**

Are there any specific types of mutations or protein interactions where the authors have observed the method to be particularly effective or challenging? Sharing such insights could help guide future research directions.

---

> ### Author Response · Authors · 2024-11-22
> **Response to Reviewer Mp9v- Part (1/4)**
>
> Dear Reviewer:
>
> Thank you for time and effort! We sincerely hope our responses fully address your questions.
>
> >**Q1**: Are there any specific types of mutations or protein interactions where the authors have observed the method to be particularly effective or challenging? Sharing such insights could help guide future research directions.
>
> **A**: Thanks for your constructive suggestion. While we have explored this, we have not yet obtained conclusive results. Nevertheless, we are happy to share our detailed analysis results.
>
> **Analysis of different protein interaction types**:
>
> | Type | P-Pea | P-Spe |
> |---------------|---------|---------|
> | AB/AG | 0.5324 | 0.5388 |
> | Pr/PI | 0.6852 | 0.6720 |
> | TCR/pMHC | 0.5941 | 0.5495 |
>
> The protein interaction type annotations in the SKEMPIv2 dataset are incomplete, with approximately 3000 data points labeled, with one-third for antibody-antigen (AB/AG), enzyme-substrate (Pr/PI), and TCR-pMHC interactions, respectively. Here we report the average per-structure correlation metric for each protein interaction type, as shown in the table. Differences in metrics across types are observed, but we cannot rule out the impact of data distribution differences.
>
> **Analysis of different mutation types**:
>  We report the frequencies of different mutation types for single-point mutations in the SKEMPIv2 dataset, along with the correlation metrics of BA-DDG, calculated separately for each mutation type.
> The rows represent the wild-type amino acid, and the columns represent the mutant amino acid.  Mutation types with frequencies greater than 10 are highlighted in **bold**.(When the frequency is too low, the correlation metric may be either uncalculable or meaningless.)
> Due to uneven data distribution, we cannot determine whether BA-DDG's lower performance on specific mutation types is due to a lack of training samples or the inherent difficulty of these types.
>
> Thanks for your recognition of our work. If you have any further questions, please feel free to raise them during the discussion phase.

---

> ### Author Response · Authors · 2024-11-22
> **Response to Reviewer Mp9v- Part (2/4)**
>
> #### Table R1. Frequencies of different mutation types.
> | | A | C | D | E | F | G | H | I | K | L | M | N | P | Q | R | S | T | V | W | Y|
> |-------------|--------|----|-------|-------|-------|-------|-------|-------|-------|-------|-------|-------|-------|-------|-------|-------|----|-------|-------|-------|
> | A | 0 | 4 | 6 | **11** | **10** | **36** | **54** | 6 | **13** | 8 | **10** | 8 | **11** | 4 | **11** | **17** | 9 | **18** | 7 | 7|
> | C | 7 | 0 | 0 | 0 | 0 | 0 | 0 | 0 | 0 | 0 | 0 | 0 | 0 | 0 | 1 | 1 | 0 | 1 | 0 | 0|
> | D | **217** | 2 | 0 | **16** | 3 | **10** | 0 | 1 | **21** | 2 | 1 | **19** | 0 | 2 | 9 | 4 | 1 | 2 | 5 | 3|
> | E | **263** | 5 | **11** | 0 | 5 | 6 | 5 | 4 | **27** | 5 | 7 | 5 | 5 | **16** | **15** | 6 | 4 | 5 | 5 | 5|
> | F | **127** | 0 | 3 | 7 | 0 | 1 | 0 | 3 | 1 | 3 | 2 | 0 | 0 | 0 | 2 | 3 | 0 | 3 | 9 | **10** |
> | G | **79** | 5 | 9 | 7 | 7 | 0 | 4 | 8 | 5 | **10** | 8 | 7 | 8 | 5 | 8 | **13** | 6 | **14** | 5 | 8|
> | H | **180** | 0 | 3 | 2 | 4 | 1 | 0 | 0 | 5 | 2 | 0 | 3 | 0 | 4 | 6 | 1 | 0 | 0 | 1 | 1|
> | I | **103** | 0 | 0 | 2 | 2 | 3 | 0 | 0 | 0 | 4 | 2 | 2 | 0 | 0 | 6 | 0 | 2 | 4 | 1 | 0|
> | K | **254** | 6 | **20** | **34** | 6 | 8 | 7 | 6 | 0 | 7 | **15** | **11** | 5 | **13** | **25** | 7 | 6 | 7 | 6 | 6|
> | L | **140** | 4 | **12** | **18** | 6 | 7 | 4 | 7 | **15** | 0 | 9 | 7 | 8 | 8 | 8 | 7 | 6 | 9 | **11** | 6|
> | M | **40** | 0 | 1 | 1 | 3 | 2 | 1 | 1 | 7 | 4 | 0 | 0 | 0 | 3 | 2 | 0 | 1 | 4 | 0 | 3|
> | N | **168** | 4 | **17** | 8 | 9 | 7 | 6 | 4 | **17** | 5 | 4 | 0 | 5 | **15** | **13** | 9 | 8 | 6 | 8 | 8|
> | P | **65** | 4 | 4 | 5 | 4 | 5 | 5 | 4 | 5 | 7 | 4 | 4 | 0 | 5 | 4 | 7 | 5 | 5 | 4 | 4|
> | Q | **151** | 0 | 0 | 9 | 0 | 0 | 0 | 0 | 7 | 5 | 0 | 1 | 0 | 0 | 5 | 0 | 1 | 3 | 0 | 0|
> | R | **290** | 6 | **10** | **29** | 7 | 9 | 8 | 4 | **23** | **12** | **11** | 8 | 9 | **15** | 0 | 6 | 7 | 6 | **10** | 8|
> | S | **196** | 3 | 6 | 4 | 2 | 5 | 0 | 0 | 3 | 3 | 2 | 4 | 5 | 7 | 5 | 0 | 5 | 1 | 0 | 7|
> | T | **166** | 4 | **11** | **13** | 9 | 4 | 5 | **11** | 9 | 7 | 4 | **13** | 8 | **10** | 9 | **17** | 0 | **19** | 6 | 5|
> | V | **110** | 1 | 3 | 3 | 3 | 1 | 0 | 7 | 2 | 3 | 1 | 1 | 0 | 0 | 1 | 1 | 7 | 0 | 1 | 2|
> | W | **95** | 0 | 1 | 0 | **21** | 0 | 0 | 0 | 2 | 1 | 0 | 0 | 1 | 1 | 2 | 0 | 0 | 1 | 0 | 6|
> | Y | **227** | 6 | 9 | 8 | **64** | 7 | 8 | 6 | 8 | **14** | 5 | 5 | 5 | 8 | 5 | 8 | 4 | 7 | **16** | 0|

---

> ### Author Response · Authors · 2024-11-22
> **Response to Reviewer Mp9v- Part (3/4)**
>
> #### Table R2. Pearson correlation of different mutation types.
> | | A | C | D | E | F | G | H | I | K | L | M | N | P | Q | R | S | T | V | W | Y |
> |-------------|-----------|---------|-------------|-----------|-----------|-----------|-----------|-----------|-----------|-----------|-----------|------------|-----------|------------|--------------|-----------|----------|-----------|-----------|----------|
> | A | | -0.1988 | 0.92 | **0.7389** | **0.7809** | **0.7848** | **0.8032** | 0.5331 | **0.916** | 0.6767 | **0.6498** | 0.746 | **0.4222** | -0.9089 | **0.8238** | **0.6467** | 0.6365 | **0.4** | -0.179 | 0.523 |
> | C | 0.3257 | | | | | | | | | | | | | | | | | | | |
> | D | **0.6937** | | | **0.5885** | 0.996 | **-0.156** | | | **0.8849** | | | **-0.0133** | | | 0.482 | 0.7649 | | | 0.4197 | 0.9998 |
> | E | **0.3422** | -0.4474 | **0.2951** | | -0.153 | 0.8348 | 0.4737 | -0.6896 | **0.9063** | 0.9175 | 0.2662 | 0.5033 | 0.8222 | **0.5973** | **0.6426** | -0.27 | -0.5772 | 0.2749 | 0.07094 | -0.503 |
> | F | **0.5839** | | 0.2144 | -0.2277 | | | | 0.1652 | | 0.994 | | | | | | 0.9442 | | 0.9171 | 0.3822 | **0.802** |
> | G | **0.5248** | 0.4917 | 0.692 | 0.733 | 0.6247 | | 0.3339 | 0.9242 | 0.9746 | **0.8853** | 0.8506 | 0.5467 | 0.2184 | 0.9244 | 0.5754 | **0.6391** | 0.4588 | **0.65** | 0.5939 | 0.4164 |
> | H | **0.8011** | | 0.7748 | | 0.816 | | | | 0.9723 | | | -0.8258 | | 0.921 | 0.5883 | | | | | |
> | I | **0.5222** | | | | | 0.8543 | | | | 0.9954 | | | | | 0.4245 | | | 0.9887 | | |
> | K | **0.7319** | -0.382 | **0.9114** | **0.8642** | 0.9647 | 0.9017 | 0.9526 | 0.9575 | | 0.9738 | **0.8621** | **0.8036** | 0.398 | **0.9556** | **-0.007102** | 0.9072 | 0.9691 | 0.9692 | 0.9935 | 0.9876 |
> | L | **0.6047** | 0.3349 | **0.8422** | **0.7506** | 0.6447 | 0.6146 | 0.6977 | 0.4082 | **0.8663** | | 0.9243 | 0.9399 | 0.9475 | 0.9666 | 0.8664 | 0.7871 | 0.7872 | 0.3547 | **0.3472** | 0.4514 |
> | M | **0.5713** | | | | 0.8778 | | | | 0.9814 | 0.8505 | | | | 0.773 | | | | -0.6849 | | 0.4637 |
> | N | **0.5415** | 0.7367 | **0.8669** | 0.8569 | 0.04623 | 0.09854 | -0.008894 | 0.0518 | **0.2815** | 0.9353 | -0.7847 | | 0.5664 | **-0.1139** | **0.7104** | 0.4796 | -0.2566 | 0.9399 | -0.02546 | -0.08176 |
> | P | **0.5458** | 0.3119 | -0.3881 | -0.05842 | 0.732 | -0.05988 | 0.6129 | 0.751 | 0.16 | 0.9591 | 0.7768 | -0.2584 | | -0.2969 | 0.8046 | 0.7951 | -0.245 | 0.8432 | 0.4119 | 0.6894 |
> | Q | **0.4618** | | | 0.4923 | | | | | 0.9333 | 0.965 | | | | | -0.6764 | | | -0.998 | | |
> | R | **0.3657** | 0.1696 | **-0.01198** | **0.3884** | 0.249 | 0.1796 | -0.3426 | -0.5455 | **0.63** | **0.5761** | **-0.127** | -0.3636 | 0.9129 | **0.3441** | | -0.2248 | 0.03203 | -0.2689 | **-0.224** | 0.3185 |
> | S | **0.4234** | 0.9938 | 0.2287 | 0.9573 | | 0.194 | | | 0.8974 | 0.9998 | | 0.2271 | 0.5408 | 0.8745 | 0.8766 | | 0.8305 | | | 0.7575 |
> | T | **0.6281** | -0.5037 | **0.8709** | **0.7617** | 0.3854 | -0.5595 | -0.1212 | **0.2019** | 0.9673 | 0.7394 | -0.6015 | **0.2592** | 0.6428 | **0.7142** | 0.7361 | **0.5646** | | **0.4604** | 0.7976 | 0.03427 |
> | V | **0.4956** | | 0.9978 | -0.01766 | 0.9583 | | | 0.6655 | | 1.0 | | | | | | | 0.6282 | | | |
> | W | **0.3589** | | | | **0.1878** | | | | | | | | | | | | | | | 0.2381 |
> | Y | **0.6807** | 0.5141 | 0.3881 | 0.7772 | **0.3131** | 0.8236 | -0.8873 | 0.5366 | 0.9026 | **0.5815** | 0.3493 | -0.5884 | 0.8759 | 0.5435 | 0.685 | 0.7062 | -0.9326 | 0.8991 | **0.4674** | |

---

> ### Author Response · Authors · 2024-11-22
> **Response to Reviewer Mp9v- Part (4/4)**
>
> #### Table R3. Spearman correlation of different mutation types.
> | | A | C | D | E | F | G | H | I | K | L | M | N | P | Q | R | S | T | V | W | Y |
> |-------------|-----------|----------|-----------|-----------|-----------|------------|-----------|-----------|------------|-----------|------------|-------------|-----------|-----------|-----------|-----------|----------|-----------|-------------|-----------|
> | A | | -0.4 | 0.9429 | **0.8182** | **0.7455** | **0.4399** | **0.6683** | 0.6 | **0.8791** | 0.6905 | **0.4061** | 0.619 | **0.5727** | -1.0 | **0.8727** | **0.336** | 0.2167 | **0.4551** | -0.1429 | 0.2857 |
> | C | -0.01802 | | | | | | | | | | | | | | | | | | | |
> | D | **0.6035** | | | **0.5294** | 1.0 | **-0.2432** | | | **0.8009** | | | **-0.02896** | | | 0.4667 | 1.0 | | | 0.6 | 1.0 |
> | E | **0.444** | -0.6 | **0.3554** | | -0.5 | 0.6 | 0.4 | -0.8 | **0.7056** | 0.5 | 0.5714 | 0.4 | 0.7 | **0.4971** | **0.5357** | -0.1429 | -0.4 | 0.4 | -0.2 | -0.6 |
> | F | **0.5885** | | -0.5 | 0.07143 | | | | -0.5 | | 0.5 | | | | | | 0.5 | | 0.5 | 0.7851 | **0.6079** |
> | G | **0.5448** | 0.7 | 0.5439 | 0.6071 | 0.3214 | | 0.0 | 0.881 | 0.7 | **0.8303** | 0.7545 | 0.3571 | 0.2619 | 0.9 | 0.7143 | **0.3736** | 0.7714 | **0.6747** | 0.8 | 0.4671 |
> | H | **0.5689** | | 1.0 | | 1.0 | | | | 0.9 | | | -1.0 | | 0.8 | 0.6 | | | | | |
> | I | **0.6039** | | | | | 0.5 | | | | 1.0 | | | | | 0.2 | | | 0.8 | | |
> | K | **0.5304** | -0.3143 | **0.4836** | **0.6981** | 0.9429 | 0.6905 | 0.3929 | 0.6 | | 0.9643 | **0.9** | **0.03636** | 0.3 | **0.5275** | **0.1581** | 0.6429 | 0.8286 | 0.4286 | 0.8286 | 0.9429 |
> | L | **0.6934** | 0.4 | **0.8322** | **0.6821** | 0.4286 | 0.75 | 0.8 | 0.1071 | **0.7536** | | 0.9624 | 0.8929 | 0.7857 | 0.8333 | 0.8095 | 0.5357 | 0.9429 | 0.5 | **0.2636** | 0.6 |
> | M | **0.4958** | | | | 0.5 | | | | 0.7857 | 0.8 | | | | 1.0 | | | | -0.8 | | 0.5 |
> | N | **0.4844** | 0.8 | **0.6691** | 0.8571 | -0.08333 | 0.07143 | -0.02857 | 0.2 | **0.02451** | 0.7 | -0.8 | | 0.6 | **0.1376** | **0.6703** | 0.1667 | -0.1317 | 0.9429 | 0.2619 | 0.1198 |
> | P | **0.5581** | 0 | -0.2 | 0.1 | 0.4 | -0.4 | 0.6 | 0.8 | -0.3 | 0.8571 | 0.8 | -0.2 | | -0.2 | 0.4 | 0.7143 | 0 | 0.9 | 0.0 | 0.4 |
> | Q | **0.4411** | | | 0.6611 | | | | | 0.5357 | 0.8 | | | | | 0.0 | | | -1.0 | | |
> | R | **0.4731** | -0.3714 | **0.1273** | **0.4246** | 0.1071 | 0.4667 | -0.04762 | -0.8 | **0.7569** | **0.3217** | **-0.1727** | -0.381 | 0.6667 | **0.5643** | | -0.08571 | 0.03571 | -0.4286 | **-0.06667** | 0.2381 |
> | S | **0.3694** | 0.866 | 0.2571 | 1.0 | | 0.3 | | | 0.5 | 1.0 | | 0.0 | 0.7 | 0.9286 | 0.9 | | 0.8 | | | 0.7857 |
> | T | **0.6356** | -0.8 | **0.8909** | **0.5165** | 0.45 | -0.6 | -0.1 | **0.3455** | 0.8667 | 0.4286 | -0.4 | **0.3516** | 0.5238 | **0.7455** | 0.7167 | **0.569** | | **0.6439** | 0.4857 | -0.1 |
> | V | **0.4676** | | 1.0 | 0.5 | 0.866 | | | 0.25 | | 1.0 | | | | | | | 0.2857 | | | |
> | W | **0.395** | | | | **0.3332** | | | | | | | | | | | | | | | -0.08571 |
> | Y | **0.644** | 0.02857 | 0.1 | 0.7857 | **0.381** | 0.8214 | -0.6667 | 0.2571 | 0.9048 | **0.4549** | 0.1 | -0.6 | 0.3 | 0.6429 | 0.3 | 0.7143 | -1 | 0.6429 | **0.5** | |

---

> ### Author Response · Authors · 2024-11-30
>
> We sincerely appreciate your time and effort in reviewing our submission! With the discussion period ending soon on Dec. 2nd, we kindly welcome any further comments or questions you may wish to share. We are happy to engage in any further opportunities for clarification.

---

> > ### Comment · Reviewer_Mp9v · 2024-12-03
> >
> > Thanks for the authors' responses and they address my concerns well. I will keep my positive score.

---

### Official Review · Reviewer_XU4j · 2024-11-03

**Soundness:** 3
**Presentation:** 4
**Contribution:** 3
**Rating:** 10
**Confidence:** 3

**Summary:**

In this paper authors propose a novel, thermodynamics inspired, approach for predicting binding free energy change upon mutation (ddG) building, in particular, on the capabilities of protein inverse folding models. Validation of the method employs standard literature benchmark (SKEMPI v2 curating and integrating a vast number of experimental measurements of ddG across various types of proteins). Beyond that, authors demonstrate applicability of their method across several tasks (binding energy prediction, antibody optimisation and docking). The reported results, in general, significantly outperform a spectrum of modern methods presented in the relevant literature both in supervised and unsupervised context.

**Strengths:**

The paper is very well written and, I believe, easy to follow both for machine-learning and also, to large extent, for biology-oriented audiences thus extending its potential impact. The limitations and weaknesses are appropriately discussed. It focuses on relevant and important problem in protein science - predicting the energetic effect of mutation on the free energy of protein-protein complex formation. The contribution is clearly original, correctly references and builds upon several already existing contributions in the field (in particular - inverse folding models, direct preference optimisation). Most importantly, benchmark reports significant improvement across several investigated metrics on commonly-accepted dataset (SKEMPI) and described ablations and controls (folded structures) strengthen the confidence in the reported result.

**Weaknesses:**

- Method is structure based, it requires at least experimentally resolved or confidently predicted complex structure,
- The representation of protein is rather coarse, only backbone. Side-chain atoms and solvent molecules are omitted yet they are obviously important drivers of protein-protein interactions.
- No representation of dynamics (structures are represented as static snapshots), which can affect the method performance for mutations inducing significant conformational changes.
- The described method seems to excel at point mutations and there's a trend towards decreased performance in more complex multiple mutations.
Majority of those weaknesses are highlighted and discussed by the authors in the appropriate sections across the paper.

**Questions:**

- In section 4.4.3 authors utilise dataset from Shen et al (2022) however focus only on point mutations. This dataset is, I think, much richer and contains double and triple mutants from multiple optimisation rounds. I believe it would be important to expand this section to show proposed model performance more extensively. This, I believe, has a potential to boost the impact and trust in the model in a typical antibody design scenario.
- Following on sec 4.4.3 - preference probabilities are a good metric but they in my view they are a bit abstract. I think a simpler metric of e.g. how the method ranks particular mutation would be more interpretable (as they relate to simple consideration of how many designs one can screen in the wet lab before finding a particular hit).
- In the typical design scenario, as the problem is hard and the reported performance of the method is not perfect, one can also consider the uncertainty of the prediction which is an important and frequently used feature in e.g. protein folding models. It could potentially strengthen the paper if authors could discuss and provide more information on how uncertainty estimation could be introduced to the model.
- In appendix B - authors write "We compare BA-DDG with several representative methods under single-point, multi-point, and
all-point mutations on SKEMPI v2". It's not clear to me what is the difference between multi-point and all-point mutations. Is the all-point a union of point- and multi-point mutations (I'd assume so by looking at the following Table) ? For clarity authors can consider to explain this more in the paper.
- The discussion about computational (non-ML) methods for ddG estimation is scarce. Even though it's purely ML venue, I miss a bit more thorough discussion of how the proposed method relates to force-field methods utilising more nuanced thermodynamical cycles / approaches like MM-GBSA/PBSA, FEP, TI, etc.

---

> ### Author Response · Authors · 2024-11-22
> **Response to Reviewer XU4j - Part (1/3)**
>
> Dear Reviewer:
>
> Thank you for your constructive comments and kind support! All your concerns have been carefully addressed as below. We sincerely hope our responses fully address your questions.
>
> >**W1**: Method is structure based, it requires at least experimentally resolved or confidently predicted complex structure.
>
> **A**: Thanks. Similar to previous state-of-the-art methods on SKEMPIv2, our approach is also structure-based. With recent advances in robust structure prediction methods, we are able to reduce our reliance on crystal structures, thus enhancing the applicability of our approach.
>
> >**W2**: The representation of protein is rather coarse, only backbone. **Side-chain atoms and solvent molecules** are omitted yet they are obviously important drivers of protein-protein interactions.
>
> Thanks. We agree that side-chain atoms and solvent molecules are important drivers of protein-protein interactions.
> Our framework can be further extended to consider side-chain conformations. In Equation (1), the probabilities can be represented as $p(X, R | S)$ instead of $p(X | S)$, where $R$ represents side-chain conformations. According to the chain rule of probability, this can be expanded as:  $p(X, R | S) = p(R | X, S) \cdot p(X | S).$ Based on this, Equation (8) can be modified to jointly leverage the probabilities on backbone and side-chain conformations to estimate $\Delta\Delta G$.
> We employ RDE to estimate the side-chain probability $p(R | X, S)$, here are the results:
>
> | Method | P-Pea | P-Spe | A-Pea | A-Spe | A-RMSE | A-MAE | A-AUROC |
> |-------|--------|--------|--------|--------|--------|--------|--------|
> | BA-DDG | 0.5453 | 0.5134 | 0.7118 | 0.6346 | 1.4516 | 1.0151 | 0.7726 |
> |BA-DDG w RDE| 0.5310 | 0.5003 | 0.7329 | 0.6486 | 1.4060 | 0.9894 | 0.7838 |
>
> The results show that BA-DDG with RDE achieves improvements on overall metrics but shows a slight decrease in per-structure metrics. This outcome may stem from challenges in side-chain conformation modeling or limitations in our current implementation. While we acknowledge the reviewer's insightful comment, we have not yet made progress in addressing these challenges, highlighting this as an important direction for future work.
>
> >**W3**: No representation of dynamics (structures are represented as static snapshots), which can affect the method performance for mutations inducing significant conformational changes.
>
> **A**: Thanks for pointing this out. We agree that for mutations inducing significant conformational changes, simply using the wild-type structure as the mutant structure may affect the results. Exploring the representation of dynamics could be an important direction for future work.
>
> >**W4**: The described method seems to excel at point mutations and there's a trend towards decreased performance in more complex multiple mutations. Majority of those weaknesses are highlighted and discussed by the authors in the appropriate sections across the paper.
>
> **A**: Thanks. As reported in Table 6, BA-DDG performs better on single-point mutations compared to multi-point mutations, which is also observed in other methods. This could be due to multi-point mutations being inherently more challenging, or it may indicate potential areas for improvement in our approach.

---

> ### Author Response · Authors · 2024-11-22
> **Response to Reviewer XU4j - Part (2/3)**
>
> >**Q1**: In section 4.4.3 authors utilise dataset from Shen et al (2022) however focus only on point mutations. This dataset is, I think, much richer and contains double and triple mutants from multiple optimisation rounds. I believe it would be important to expand this section to show proposed model performance more extensively. This, I believe, has a potential to boost the impact and trust in the model in a typical antibody design scenario.
>
> **A**: Thanks for your suggestion. Results for multi-point mutations are as follows, using the same setting as described in Section 4.4.3.
>
> | mutations | perplexity (BA-DDG) | perplexity (ProteinMPNN) | preference (BA-DDG) | preference (ProteinMPNN) |
> |--------|-------------------|------------------------|-------------------|----------------------|
> | RH103M,LH104F | 1.556 | 2.524 | 32.12% | 17.60% |
> | TH30P,RH103M | 1.023 | 1.310 | 53.22% | 45.09% |
> | TH31W,RH103M | 1.669 | 3.894 | 36.87% | 13.63% |
> | AH39F,RH103M | 1.658 | 6.312 | 31.13% | 2.478% |
> | NH55L,RH103M | 1.465 | 2.600 | 36.04% | 15.20% |
> | NH57L,RH103M | 1.366 | 0.936 | 44.32% | 57.68% |
> | TH30P,RH103M,LH104F | 1.151 | 2.181 | 45.21% | 22.29% |
> | TH31W,RH103M,LH104F | 1.442 | 2.941 | 39.40% | 22.71% |
> | TH30P,TH31W,RH103M | 1.203 | 2.606 | 45.74% | 22.44% |
> | AH39F,RH103M,LH104F | 1.300 | 4.621 | 37.94% | 4.705% |
> | AH39F,NH55L,RH103M | 1.500 | 4.929 | 34.63% | 3.416% |
> | NH55L,NH57L,RH103M | 1.210 | 1.403 | 42.93% | 36.75% |
> | AH39F,NH57L,RH103M | 1.409 | 2.329 | 38.18% | 15.82% |
> | TH30P,AH39F,RH103M | 1.136 | 2.841 | 43.07% | 10.82% |
> | TH31W,AH39F,RH103M | 1.606 | 6.742 | 34.08% | 2.438% |
> | TH31W,NH55L,RH103M | 1.419 | 3.302 | 39.33% | 18.19% |
> | TH30P,NH57L,RH103M | 1.012 | 0.948 | 52.97% | 51.40% |
> | TH31W,NH57L,RH103M | 1.465 | 2.556 | 40.62% | 27.91% |
> | AH39F,TH30P,RH103M,LH104F | 1.081 | 3.036 | 44.56% | 7.513% |
> | Average | **1.351** | 3.053 | **40.65%** | 20.95% |
>
>
> >**Q2**: Following on sec 4.4.3 - preference probabilities are a good metric but they in my view they are a bit abstract. I think a simpler metric of e.g. how the method ranks particular mutation would be more interpretable (as they relate to simple consideration of how many designs one can screen in the wet lab before finding a particular hit).
>
> **A**:  Thanks for your suggestion. For a CDR region of length $L$ with $K$ mutations, the number of possible mutants is $C(L, K) \times 19^K$. For double and triple mutations, this results in 117,325 and 17,833,400 mutants, respectively. Here we report the ranking results only for single-point mutations.
> | Method | TH31W | AH53F | NH57L | RH103M | LH104F | Average |
> |----------|-----------|-----------|-----------|------------|------------|-------------|
> | ProteinMPNN | 32.59% | 72.47% | 2.23% | 78.95% | 38.66% | 44.18% |
> | BA-DDG | 3.64% | 3.04% | 12.15% | 70.04% | 12.55% | 20.24% |

---

> ### Author Response · Authors · 2024-11-22
> **Response to Reviewer XU4j - Part (3/3)**
>
> >**Q3**: In the typical design scenario, as the problem is hard and the reported performance of the method is not perfect, one can also consider the uncertainty of the prediction which is an important and frequently used feature in e.g. protein folding models. It could potentially strengthen the paper if authors could discuss and provide more information on how **uncertainty estimation** could be introduced to the model.
>
> **A**: Thank you for this insightful comment. We agree that uncertainty estimation is an important yet challenging problem. While metrics such as pLDDT in protein folding models provide valuable confidence scores, they are also known to suffer from false positives. Extending uncertainty estimation to our approach would require careful consideration of similar issues, and we see this as an interesting direction for future work.
> We have considered three possible directions for introducing uncertainty estimation:
> 1.  **Inference Ensembles:** Using multiple models or parameter sets during inference to assess the variability in predictions.
> 2.  **Dropout as a Bayesian Approximation:** Applying dropout during inference to approximate a posterior distribution over model predictions.
> 3.  **Inverse Folding Model Uncertainty:** Extending the inverse folding model to output uncertainty metrics similar to pLDDT in protein folding models. These uncertainty estimates could then be integrated into BA-DDG for uncertainty estimation of $\Delta\Delta G$.
>
> We hope to explore these ideas in our future work.
>
> >**Q4**: In appendix B - authors write "We compare BA-DDG with several representative methods under single-point, multi-point, and all-point mutations on SKEMPI v2". It's not clear to me what is the difference between multi-point and all-point mutations. Is the all-point a union of point- and multi-point mutations (I'd assume so by looking at the following Table) ? For clarity authors can consider to explain this more in the paper.
>
> **A**: Thanks and addressed. Your understanding is correct: "all-point" indeed represents a union of point- and multi-point mutations. We will clarify this distinction in the revised version of the paper to improve clarity.
>
> >**Q5**: The discussion about computational (non-ML) methods for ddG estimation is scarce. Even though it's purely ML venue, I miss a bit more thorough discussion of how the proposed method relates to force-field methods utilising more nuanced thermodynamical cycles / approaches like MM-GBSA/PBSA, FEP, TI, etc.
>
> **A**: As we are less familiar with these methods, and given that FEP can be computationally expensive, we are still working on this. We hope to provide some reliable results by Nov 26.
>
> **Update (Nov 27)**:
> We have attempted to conduct $\Delta\Delta G$ estimations using non-ML methods, including MM-PBSA and FEP, on a subset of the SKEMPIv2 dataset. However, the results were lower than expected, likely due to our limited experience with these non-ML methods. We are continuing our efforts and will share reliable results if obtained before the discussion period ends.

---

> > ### Comment · Reviewer_XU4j · 2024-11-29
> >
> > I'd like to thank the authors for all the effort in providing the additional convincing results. In sum, taking into the account also the suggestions and concerns raised by other reviewers and authors' responses, I believe it's a strong and novel work that is thoroughly benchmarked and with clearly noted limitations. As such, I think it should be highlighted, therefore I find it appropriate to raise my score accordingly.

---

> > > ### Author Response · Authors · 2024-11-30
> > >
> > > We sincerely appreciate your valuable suggestions and kind encouragement! With the discussion period ending on Dec. 2nd, we welcome any further questions you may have.

---

### Official Review · Reviewer_V65x · 2024-11-05

**Soundness:** 3
**Presentation:** 3
**Contribution:** 3
**Rating:** 6
**Confidence:** 3

**Summary:**

This paper employs pre-trained inverse folding models based on the Boltzmann distribution and thermodynamic cycles to associate energy with log-likelihood from inverse folding models for predicting ΔΔG. It introduces both unsupervised and supervised inverse folding models, each achieving state-of-the-art (SoTA) accuracy on the SKEMPI v2 dataset. Additionally, the study demonstrates that the proposed thermodynamic cycle is effective and offers advantages over traditional SFT and DPO methods. Notably, the method does not strictly require crystal structures when a reasonably accurate predicted structure is available. Furthermore, the paper explores applications of this approach in binding affinity energy prediction, protein-protein docking, and antibody optimization.

**Strengths:**

1. It creatively used Boltzmann Alignment, which introduces physical inductive bias through Boltzmann distribution and thermodynamic cycle to link binding energy and log-likelihood in inverse folding models.

2. This work achieved SOTA and outperforms other SFT and DPO methods. The method can not only be used in binding energy prediction, but also be used in  protein-protein docking, and antibody optimization.

**Weaknesses:**

1. The code is not available,  it should be open-sourced upon publication.

2. Dividing the dataset into 3 folds by structure cannot strictly prevent data leakage, different protein complexes might have similar protein binding structures.

**Questions:**

1. In Section 4.1, Dividing the dataset into 3 folds by structure cannot strictly prevent data leakage, different protein complexes might have similar protein binding structures, can you elaborate more about the method you used to split data and prevent data leakage?
Could you explain more about data partition process, why it can help to reduce data leakage process?

2. Could you explain more about data partition process, and why it can help to reduce data leakage process?

3. The SKEMPI dataset includes both single-point mutations and multiple-point mutations. Given the varying complexity, predicting binding affinity for single-point mutations may be less challenging and could potentially yield more accurate prediction results compared to multiple-point mutations. This raises the question: why not consider training separate models tailored for single and multiple mutation binding affinity prediction tasks, rather than using a single model for both?

---

> ### Author Response · Authors · 2024-11-22
> **Response to Reviewer V65x**
>
> Dear Reviewer:
>
> Thank you for your thorough review and insightful comments! In the following, we will address each issue in detail and provide the necessary experimental results.
>
> >**W1**: The code is not available, it should be **open-sourced** upon publication.
>
> **A**: Thanks for your suggestion. We are currently organizing and refactoring the code, and we plan to **release our code upon publication**.
>
> >**W2 & Q1 & Q2**: In Section 4.1, Dividing the dataset into 3 folds by structure **cannot strictly prevent data leakage**, different protein complexes might have similar protein binding structures, can you elaborate more about the method you used to split data and prevent data leakage? Could you explain more about data partition process, why it can help to reduce data leakage process?
>
> **A**: Thanks for your insightful comment.
> (1) Our decision to randomly divide the dataset into 3 folds by structure was primarily to **ensure consistency and fair comparison with previous works**, including RDE, Surface-VQMAE, DDGPred, Prompt-DDG, DiffAffinity and ProMIM.
> (2) From the perspective of $\Delta\Delta G$ labels, structure-based split does not introduce data leakage.
> (3) We have the same concern with you that different protein complexes might share **similar protein binding structures**. A more rigorous approach, such as clustering structures based on TM-scores, could be considered in the future. However, this has not yet been explored for SKEMPI v2 in existing studies.
>
>
> >**Q3**: The SKEMPI dataset includes both single-point mutations and multiple-point mutations. Given the varying complexity, predicting binding affinity for single-point mutations may be less challenging and could potentially yield more accurate prediction results compared to multiple-point mutations. This raises the question: why not **consider training separate models** tailored for single and multiple mutation binding affinity prediction tasks, rather than using a single model for both?
>
> **A**:   Thanks for your insightful suggestion. We explored training separate models for single- and multiple-point mutations but observed no significant improvement in performance compared to a unified model under the same training conditions. We speculate that, despite the increased complexity of multiple-point mutations, both tasks fundamentally address the same underlying problem, and leveraging shared data does not seem to harm performance.
>
> | Model | Mutation | P-Pea | P-Spe | A-Pea | A-Spe | A-RMSE | A-MAE | A-AUROC |
>  |---------|----------|--------|--------|--------|--------|--------|--------|--------|
>  | unified | single | 0.5606 | 0.5192 | 0.7321 | 0.6157 | 1.1848 | 0.8409 | 0.7662 |
> | separate | single | 0.5503 | 0.5176 | 0.7277 | 0.6078 | 1.1929 | 0.8458 | 0.7693 |
> | unified | multiple | 0.4924 | 0.4959 | 0.6650 | 0.6293 | 2.0151 | 1.4944 | 0.7875 |
> | separate | multiple | 0.4869 | 0.4547 | 0.6928 | 0.6101 | 1.9457 | 1.4878 | 0.7934 |

---

> ### Author Response · Authors · 2024-11-30
>
> Thank you for your insightful comments once again! As the discussion period is coming to close (Dec. 2nd), if you have any further questions, please feel free to raise them. We are happy to engage in any further opportunities for clarification.

---

### Meta-Review · Area_Chair_kRVT · 2024-12-17

**Metareview:**

The paper considers protein-protein interactions and proposes Boltzmann Alignment to predict the change in binding free energy resulting from mutation. The approach make use of physical inductive bias through Boltzmann distribution and thermodynamic cycle to link pre-trained inverse folding models to binding free-energy prediction. The paper is a pleasure to read and accessible. The contributions are novel and the approach intuitive and interesting. The value of the proposed method achieves SOTA results on SKEMPI v2 benchmark and its broader applicability is also convincingly demonstrated for binding energy prediction, protein-protein docking, and antibody optimization.

**Additional Comments On Reviewer Discussion:**

The authors have done a great job at addressing the reviewers concerns and have provided valuable additional experimental results, including training of separate models for single-point vs multiple-point mutations. The additional results on multi-point mutations reported to reviewer XU4j are impressive and it would be great to include them in the supplements.  We also encourage the authors to incorporate the remark they provided to Reviewer 5mfi, justifying the assumption to treat the unbound state as independent probabilities.

All reviewers are satisfied by the author responses and have maintained or raised their scores as a result. The AC fully agrees that this is a strong paper.

---

### Decision · Program_Chairs · 2025-01-22

Accept (Spotlight)